# DOUBLY ROBUST MONTE CARLO TREE SEARCH

## ABSTRACT

We present Doubly Robust Monte Carlo Tree Search (DR-MCTS), an algorithm that integrates doubly robust off-policy estimation into MCTS to improve sample efficiency. Our hybrid estimator combines Monte Carlo rollouts with DR estimation through a variance-minimizing weight computed online. Unlike biased bootstrapping methods that sacrifice asymptotic correctness, DR-MCTS achieves variance reduction while preserving unbiasedness. Unlike entropy-based approaches that exhibit domain-dependent performance, DR-MCTS demonstrates consistent improvements across diverse settings including game playing, mathematical reasoning, and embodied planning. The benefits are particularly pronounced in LLM-augmented environments where each simulation is computationally expensive, making DR-MCTS well-suited for the growing class of language-model-guided planning applications.

## 1 INTRODUCTION

Monte Carlo Tree Search (MCTS) has emerged as a powerful approach for sequential decision-making, demonstrating remarkable success in domains ranging from game playing (Browne et al., 2012; Silver et al., 2016) to enhancing the reasoning capabilities of Large Language Models (Yao et al., 2023; Zhou et al., 2024). Central to MCTS is the estimation of state values, where lower variance enables more reliable action selection: with high-variance estimates, the algorithm may incorrectly identify suboptimal actions as best, wasting computational budget on poor branches. This challenge intensifies as sampling costs grow, particularly in LLM applications where each node expansion may involve expensive model queries.

A key consideration in MCTS is *whose* value we estimate. Standard rollouts sample from a behavior policy (often uniform random or heuristic-guided), but we ultimately care about the value under the *target* policy—the greedy policy derived from current Q-estimates. When behavior and target policies diverge significantly, naive Monte Carlo estimates become biased proxies for the quantity of interest, potentially leading to suboptimal decisions even with many samples. Off-policy correction addresses this mismatch, ensuring our estimates reflect the policy we intend to execute.

Existing approaches to improving MCTS sample efficiency have pursued several strategies: entropy-based exploration (Xiao et al., 2019; Painter et al., 2023), alternative backup strategies (Khandelwal et al., 2016), and structural state sharing (Grosse et al., 2021). Methods like MaxMCTS reduce variance through biased value bootstrapping, but sacrifice the unbiasedness that guarantees asymptotic correctness—a trade-off that proves problematic in sparse-reward domains. Meanwhile, AlphaZero-style approaches (Silver et al., 2017) replace rollouts entirely with learned value functions, but the neural network estimates themselves exhibit variance that propagates through the search tree.

We introduce MCTS-DR, an algorithm that integrates Doubly Robust (DR) off-policy estimation into MCTS to achieve variance reduction while preserving unbiasedness. Our approach employs a hybrid estimator that combines traditional MCTS rollouts with DR estimation through a variance-minimizing weight computed online from empirical statistics. The key insight is that neither estimator dominates across all conditions: MCTS rollouts provide stable estimates but ignore policy mismatch, while DR estimation corrects for distributional shift but suffers from importance weight explosion when policies diverge significantly. By optimally combining both, MCTS-DR achieves variance below either component alone.

Our contributions are as follows:

1. We introduce MCTS-DR, the first algorithm integrating doubly robust off-policy estimation into MCTS through an adaptive hybrid estimator with variance-minimizing weights.

2. We establish theoretical guarantees proving that the hybrid estimator is unbiased for any mixing coefficient, and that the optimal weight achieves variance no greater than either component estimator. We further show how variance reduction translates to improved action selection probability and reduced sample complexity.

3. We conduct extensive evaluation across four domains with different characteristics: Go (sparse rewards, large action space), Atari (dense rewards), GSM8K mathematical reasoning (LLM policies), and VirtualHome (compositional planning). MCTS-DR achieves consistent improvements across all domains, demonstrating robustness where baselines exhibit domain-dependent performance. Notably, MCTS-DR provides complementary benefits even when combined with neural network value functions, outperforming AlphaZero-style baselines.

## 1.1 Related Work

Our work draws from two primary research streams: advances in Monte Carlo Tree Search and off-policy evaluation techniques in reinforcement learning.

**Monte Carlo Tree Search.** Since its introduction by Coulom (2006), MCTS has become a fundamental algorithm for sequential decision-making, achieving notable success across diverse domains (Silver et al., 2016; 2018; Schrittwieser et al., 2020). Recent work has pursued sample efficiency through several complementary approaches.

Entropy-based methods modify the exploration-exploitation balance during search. Xiao et al. (2019) introduced Maximum Entropy Monte-Carlo Planning (MENTS), achieving exponential convergence rates through softmax backpropagation, though the entropy objective can conflict with reward maximization. Painter et al. (2023) addressed this with Boltzmann Tree Search (BTS) and Decaying ENtropy Tree-Search (DENTS), which gradually transitions from exploration to exploitation.

Alternative backup strategies offer another avenue for improvement. Khandelwal et al. (2016) systematically analyzed complex backup strategies from the RL literature applied to MCTS, proposing MaxMCTS variants that use $\lambda$-returns and take the maximum over Monte Carlo returns and bootstrapped value estimates. While MaxMCTS can reduce variance through biased bootstrapping, it sacrifices the unbiasedness that guarantees asymptotic correctness, a trade-off that proves problematic in sparse-reward domains where biased estimates cannot self-correct.

Structural approaches exploit similarities across the search space. Grosse et al. (2021) proposed Probabilistic DAG Search, using jointly Gaussian models to share information across states. Borges & Oliveira (2021) explored utilizing the off-policy data naturally generated during MCTS exploration.

**Off-Policy Evaluation and Doubly Robust Methods.** The doubly robust (DR) framework, originating in causal inference (Robins & Rotnitzky, 1995), addresses the high variance of importance sampling (Precup et al., 2000) by combining IS with direct value estimation. Dudík et al. (2011) first applied DR to contextual bandits; Jiang & Li (2016) extended it to sequential RL, establishing theoretical foundations. Subsequent refinements include weighted DR estimators (Thomas & Brunskill, 2016), the More Robust Doubly Robust estimator (Farajtabar et al., 2018), and double reinforcement learning (Kallus & Uehara, 2020).

**Our Contribution.** MCTS-DR represents the first direct integration of doubly robust estimation into MCTS. Unlike entropy-based methods that modify exploration strategy, or MaxMCTS which reduces variance through biased bootstrapping, we achieve variance reduction while preserving unbiasedness through our hybrid estimator with variance-minimizing weight $\beta^*$. As illustrated in Figure 1, while standard MCTS relies solely on Monte Carlo sampling during simulation, DR-MCTS employs this hybrid estimator to achieve more accurate value estimates with fewer samples.

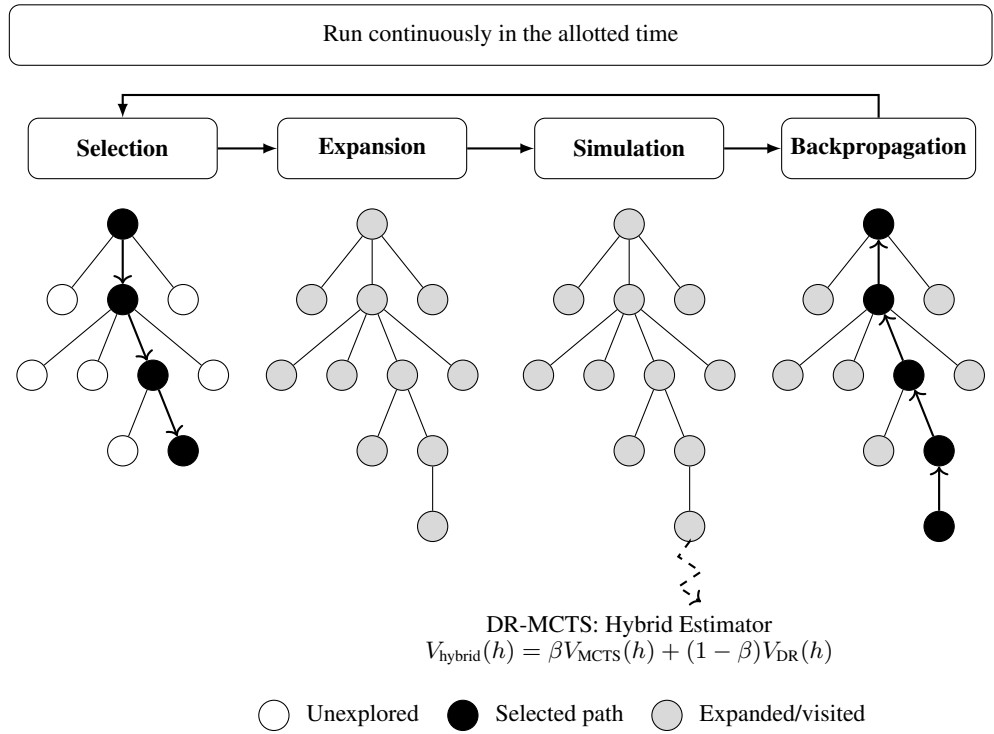

Figure 1: Monte Carlo Tree Search phases. Standard MCTS uses pure Monte Carlo rollouts in the simulation phase, while DR-MCTS employs a hybrid estimator combining MCTS rollouts with doubly robust off-policy estimation.

## 1.2 BACKGROUND

### 1.2.1 MARKOV DECISION PROCESSES

We formalize our problem setting as a Markov Decision Process (MDP), defined by the tuple $M = \langle S, A, P, R, \gamma \rangle$. Here, $S$ and $A$ represent the state and action spaces respectively, while $P : S \times A \times S \to [0, 1]$ captures the transition dynamics, with $P(s'|s, a)$ denoting the probability of transitioning to state $s'$ after taking action $a$ in state $s$. The reward function $R : S \times A \to \mathbb{R}$ assigns immediate rewards $R(s, a)$ to state-action pairs, and $\gamma \in [0, 1]$ serves as the discount factor for future rewards.

A trajectory through this MDP unfolds as a sequence $\tau = (s_0, a_0, r_0, s_1, a_1, r_1, \ldots, s_H)$, where each reward $r_t = R(s_t, a_t)$ follows from the corresponding state-action pair. An agent's behavior is governed by a policy $\pi : S \to \Delta(A)$, which maps states to probability distributions over actions. The fundamental objective is to find a policy that maximizes the expected cumulative discounted reward:

$$V^\pi(s) = \mathbb{E}_\pi \left[ \sum_{t=0}^{H} \gamma^t r_t \,\middle|\, s_0 = s \right] \tag{1}$$

Many practical domains, including board games like Go and simulated environments like VirtualHome (Puig et al., 2018), exhibit sparse reward structures where feedback occurs primarily at terminal states. This sparsity presents a significant challenge for value estimation, as intermediate actions receive no immediate signal about their quality.

### 1.2.2 MONTE CARLO TREE SEARCH

Monte Carlo Tree Search addresses the challenge of decision-making in large state spaces through selective sampling and incremental tree construction (Browne et al., 2012). Rather than exhaustively exploring all possibilities, MCTS focuses computational resources on promising regions of the search space through an iterative four-phase process.

The algorithm begins with *selection*, traversing from the root to a leaf node by balancing exploration of uncertain branches with exploitation of promising ones. We implement this trade-off using the Predictor Upper Confidence Trees (PUCT) criterion (Rosin, 2011; Silver et al., 2017):

$$a^* = \arg\max_a \left( Q(s, a) + c\pi_b(a|s)\frac{\sqrt{N(s)}}{1 + N(s, a)} \right) \tag{2}$$

This formula elegantly combines the estimated action value $Q(s, a)$ with an exploration bonus that decreases as the state-action pair $(s, a)$ is visited more frequently, where $N(s)$ and $N(s, a)$ track visit counts and $c$ controls the exploration-exploitation balance. The prior policy $\pi_b(a|s)$ biases exploration toward promising actions; in our experiments, this is either uniform (Go), derived from an LLM policy model (VirtualHome), or based on reasoning heuristics (GSM8K).

Upon reaching a leaf, the *expansion* phase adds new child nodes to grow the tree. The crucial *simulation* phase then estimates the leaf's value through Monte Carlo rollouts:

$$V_{\text{MCTS}}(s) = \frac{1}{N(s)} \sum_{i=1}^{N(s)} R_i(s) \tag{3}$$

where each $R_i(s)$ represents the cumulative reward from an independent simulation starting at state $s$. Finally, *backpropagation* updates the statistics of all nodes along the traversed path, propagating the new information toward the root.

While this Monte Carlo approach provides unbiased estimates, it can suffer from high variance, particularly when simulations are expensive or limited.

### 1.3 OFF-POLICY EVALUATION METHODS

The challenge of estimating a policy's value using data collected under a different policy arises naturally in MCTS, where the tree policy used for exploration differs from the target policy we ultimately wish to evaluate. Off-policy evaluation methods provide principled approaches to this mismatch.

#### 1.3.1 IMPORTANCE SAMPLING

Importance Sampling (IS) corrects for the distribution mismatch between behavior policy $\pi_b$ and target policy $\pi_e$ through density ratios (Precup et al., 2000). For each timestep, we compute the importance weight $\rho_t = \pi_e(a_t|s_t)/\pi_b(a_t|s_t)$, which reweights the observed data to match what would have been observed under the target policy. The step-wise IS estimator accumulates these weighted rewards:

$$V_{\text{step-IS}}(s) = \sum_{t=0}^{H-1} \gamma^t \rho_{1:t} r_t \tag{4}$$

where $\rho_{1:t} = \prod_{k=1}^{t} \rho_k$ represents the cumulative importance weight. While this approach provides unbiased estimates under mild conditions, the variance can become prohibitive when the behavior and target policies diverge significantly.

#### 1.3.2 DOUBLY ROBUST ESTIMATION

Doubly Robust estimation elegantly addresses the high-variance limitation of IS by incorporating a baseline function that reduces variance without introducing bias (Jiang & Li, 2016). The key insight is to combine importance sampling with direct value function approximation:

$$V_{\text{DR}}(s) = \hat{V}(s) + \sum_{t=0}^{H-1} \gamma^t \rho_{1:t} \left( r_t + \gamma\hat{V}(s_{t+1}) - \hat{Q}(s_t, a_t) \right) \tag{5}$$

This formulation starts with a baseline estimate $\hat{V}(s)$ and adds a correction term that accounts for the discrepancy between observed rewards and predicted values. Crucially, the estimator remains

unbiased if *either* the importance weights are correct *or* the value function approximations are accurate—hence the term "doubly robust." This robustness property makes DR estimation particularly attractive for complex domains where perfect models are unattainable.

To further reduce bias in finite-sample settings, we employ cross-validation when estimating $\hat{Q}(s_t, a_t)$ (Chernozhukov et al., 2018), preventing overfitting to the limited data available within each MCTS node. This combination of robustness and practical bias reduction techniques forms the foundation of our DR-MCTS algorithm.

## 2 METHODS

### 2.1 DOUBLY ROBUST MONTE CARLO TREE SEARCH

Our DR-MCTS algorithm enhances the standard MCTS framework by introducing a variance-minimizing hybrid estimator that adaptively combines Monte Carlo rollouts with doubly robust off-policy evaluation. The core innovation lies in dynamically adjusting the mixture weights based on empirical variance statistics, allowing the algorithm to optimally balance different sources of value information.

The hybrid estimator takes the form:

$$V_{\text{hybrid}}(s) = \beta(s, a)V_{\text{MCTS}}(s) + (1 - \beta(s, a))V_{\text{DR}}(s) \tag{6}$$

where $\beta(s, a) \in [0, 1]$ determines the relative contribution of each component. Rather than using a fixed or heuristically-decaying weight, we compute $\beta(s, a)$ online to minimize the combined estimator's variance.

The key to our approach is the adaptive computation of $\beta(s, a)$ based on observed variance statistics. For each state-action pair, we maintain online estimates of the variances and covariance of the two estimators. The variance-minimizing weight is then computed as:

$$\beta^*(s, a) = \frac{\text{Var}(V_{\text{DR}}) - \text{Cov}(V_{\text{MCTS}}, V_{\text{DR}})}{\text{Var}(V_{\text{MCTS}}) + \text{Var}(V_{\text{DR}}) - 2\text{Cov}(V_{\text{MCTS}}, V_{\text{DR}})} \tag{7}$$

where the variance and covariance terms are estimated online using a sliding window of recent samples. When insufficient data is available for reliable variance estimation (typically in the first few visits), we fall back to an exponentially decaying heuristic:

$$\beta_{\text{fallback}}(s, a) = \beta_{\text{base}} \cdot \exp(-\lambda \cdot N(s, a)) \tag{8}$$

This ensures reasonable behavior during the initial exploration phase while transitioning to optimal variance-based weighting as data accumulates.

To compute the hybrid estimator in practice, we first calculate the value function estimate $\hat{V}(s)$ using Equation 10 and the Q-value estimates $\hat{Q}(s_t, a_t)$ using Equation 11. These estimates are then substituted into the doubly robust estimator $V_{\text{DR}}(s)$ in Equation 5. Finally, the hybrid estimator $V_{\text{hybrid}}(s)$ in Equation 6 combines the standard MCTS rollout value $V_{\text{MCTS}}(s)$ with the doubly robust estimate $V_{\text{DR}}(s)$ using the adaptive weighting parameter $\beta(s, a)$.

The target policy $\pi_e$ for importance sampling is derived from current Q-value estimates using a softmax distribution:

$$\pi_e(a|s) = \frac{\exp(Q(s, a))}{\sum_{a'} \exp(Q(s, a'))} \tag{9}$$

We considered alternative target policies including $\epsilon$-greedy, visit-count-based ($\pi_e(a|s) \propto N(s, a)$), and UCB-based formulations. In ablation studies (Table 8), softmax consistently outperformed these alternatives, likely because it provides smooth probability gradients that yield more stable importance weights while still concentrating mass on high-value actions. The behavior policy $\pi_b(a|s)$ varies by domain as detailed in Appendix C.

We estimate the value function as a weighted average over child nodes:

$$\hat{V}(s) = \sum_a \pi_e(a|s) \cdot \frac{1}{N(s, a)} \sum_{i=1}^{N(s,a)} R_i(s, a) \tag{10}$$

where $R_i(s, a)$ denotes the $i$-th return observed from taking action $a$ in state $s$.

For action-value estimation, we employ k-fold cross-validation to reduce overfitting bias:

$$\hat{Q}(s_t, a_t) = \frac{1}{K} \sum_{k=1}^{K} \frac{1}{|D_k|} \sum_{i \in D_k} R_i(s_t, a_t) \tag{11}$$

where the data is partitioned into $K$ folds, with each fold $D_k$ providing an independent estimate.

The complete DR-MCTS algorithm is provided in Algorithm 1 in the Appendix.

## 2.2 THEORETICAL ANALYSIS

Our theoretical analysis establishes how variance reduction in value estimation translates to improved decision-making performance. We first present foundational properties of our hybrid estimator, then derive our main results connecting variance reduction to action selection quality and sample complexity.

**Corollary 2.1** (Unbiasedness of Hybrid Estimator)**.** *The hybrid estimator $V_{hybrid}(s) = \beta(s, a)V_{MCTS}(s) + (1 - \beta(s, a))V_{DR}(s)$ is unbiased for estimating the value of the target policy $\pi_e$ for any $\beta(s, a) \in [0, 1]$.*

*Proof Sketch.* Since both $V_{MCTS}(s)$ and $V_{DR}(s)$ are unbiased estimators of $V^{\pi_e}(s)$ (Jiang & Li, 2016), any convex combination preserves unbiasedness by linearity of expectation. See Appendix B.1 for details. □

**Corollary 2.2** (Variance-Minimizing Weight)**.** *The optimal mixing coefficient that minimizes $Var(V_{hybrid}(s))$ is:*

$$\beta^*(s, a) = \frac{Var(V_{DR}) - Cov(V_{MCTS}, V_{DR})}{Var(V_{MCTS}) + Var(V_{DR}) - 2Cov(V_{MCTS}, V_{DR})} \tag{12}$$

*and the resulting variance satisfies $Var(V_{hybrid}^*) \leq \min\{Var(V_{MCTS}), Var(V_{DR})\}$.*

*Proof Sketch.* This follows from the classical result on optimal linear combinations of correlated estimators (Graybill & Deal, 1959). The optimal weight is obtained by differentiating the variance of the convex combination with respect to $\beta$ and setting the derivative to zero. See Appendix B.2 for the full derivation. □

We now present our main contributions: theorems establishing how variance reduction leads to measurable improvements in decision quality.

**Definition 2.3** (Value Gap)**.** *For state $s$ with optimal action $a^* = \arg\max_a Q^*(s, a)$, define the value gap as $\Delta_{\min}(s) = \min_{a \neq a^*}[Q^*(s, a^*) - Q^*(s, a)]$.*

**Theorem 2.4** (Variance Reduction Improves Action Selection)**.** *Let $\hat{Q}_1(s, a)$ and $\hat{Q}_2(s, a)$ be unbiased Q-value estimators with variances $\sigma_1^2(s, a)$ and $\sigma_2^2(s, a)$ respectively, where $\sigma_2^2(s, a) \leq \sigma_1^2(s, a)$ for all $(s, a)$. Define the probability of selecting the optimal action as $P_{opt}^{(i)}(s) = P\left(\arg\max_a \hat{Q}_i(s, a) = a^*\right)$.*

*Then for any state $s$ with value gap $\Delta_{\min}(s) > 0$:*

$$P_{opt}^{(2)}(s) \geq P_{opt}^{(1)}(s) \tag{13}$$

*with strict inequality when $\sigma_2^2(s, a) < \sigma_1^2(s, a)$ for at least one action $a$.*

*Furthermore, under Gaussian approximation, the improvement is lower bounded by:*

$$P_{opt}^{(2)}(s) - P_{opt}^{(1)}(s) \geq \Phi\left(\frac{\Delta_{\min}(s)}{\sqrt{2}\sigma_2}\right) - \Phi\left(\frac{\Delta_{\min}(s)}{\sqrt{2}\sigma_1}\right) \tag{14}$$

*where $\Phi(\cdot)$ is the standard normal CDF and $\sigma_i = \max_a \sigma_i(s, a)$.*

*Proof Sketch.* The key insight is that selecting the optimal action requires correctly ranking $\hat{Q}(s, a^*)$ above all suboptimal actions. For unbiased estimators, the probability of misranking two actions is bounded by the ratio of estimation variance to the squared value gap (via Chebyshev's inequality). Lower variance directly reduces misranking probability. The Gaussian bound follows from the CLT applied to averaged rollout returns. See Appendix B.3 for the complete proof. $\square$

**Remark 2.5.** *This result connects to the best-arm identification literature (Kaufmann et al., 2016), where sample complexity depends on the gap between arm means and the estimation variance. Our contribution is showing this relationship holds within MCTS when comparing estimators of different variance.*

**Theorem 2.6** (Sample Complexity Improvement). *Consider an MCTS algorithm requiring $N_1$ simulations to achieve $P_{opt}(s) \geq 1 - \delta$ for selecting the optimal action. Let DR-MCTS achieve variance reduction factor $\rho = Var(V_{hybrid})/Var(V_{MCTS}) < 1$. Then DR-MCTS achieves the same guarantee with at most:*

$$N_2 \leq \rho \cdot N_1 \tag{15}$$

*simulations.*

*More precisely, to achieve optimal action selection probability $1 - \delta$ in a state with value gap $\Delta_{\min}$, MCTS requires:*

$$N_{MCTS} = O\left( \frac{\sigma^2_{rollout}}{\Delta^2_{\min}} \log \frac{|A|}{\delta} \right) \tag{16}$$

*while DR-MCTS requires:*

$$N_{DR\text{-}MCTS} = O\left( \frac{\rho \cdot \sigma^2_{rollout}}{\Delta^2_{\min}} \log \frac{|A|}{\delta} \right) \tag{17}$$

*Proof Sketch.* By the CLT, MCTS Q-value estimation variance scales as $\sigma^2_{\text{rollout}}/N$. To ensure the optimal action is selected with probability $\geq 1 - \delta$, the estimation error must be smaller than half the value gap with high probability. Applying Hoeffding's inequality with a union bound over $|A|$ actions yields the stated sample complexity. Since DR-MCTS achieves variance $\rho \cdot Var(V_{\text{MCTS}})$ with $\rho < 1$, the sample complexity reduces proportionally. See Appendix B.4 for details. $\square$

**Remark 2.7.** *The $O(\sigma^2/\Delta^2 \cdot \log(|A|/\delta))$ form matches the PAC sample complexity bounds for best-arm identification in multi-armed bandits (Even-Dar et al., 2006; Kaufmann et al., 2016), confirming that our bounds are tight up to constants.*

## 3 EXPERIMENTS AND RESULTS

We evaluate MCTS-DR across four domains spanning different reward structures, action spaces, and planning horizons: Go (sparse rewards, large branching factor), Atari (dense rewards), GSM8K mathematical reasoning (LLM-based policies), and VirtualHome household tasks (compositional planning). Detailed domain descriptions are provided in Appendix **??**, with hyperparameter configurations in Table 5.

Our evaluation compares MCTS-DR against baselines representing different approaches to value estimation and exploration. Table 4 summarizes which baselines are evaluated in each domain.

**Core baselines (all domains). MCTS** with pure Monte Carlo rollouts serves as the primary baseline. **DR** uses only the model-based DR estimator without MCTS rollouts, isolating the contribution of importance-weighted corrections. **MCTS-IS** replaces DR with step-wise importance sampling (Equation 4) in our hybrid formulation, enabling direct comparison between DR and IS corrections.

**Variance reduction and exploration baselines (Go, Atari, GSM8K). MaxMCTS** (Khandelwal et al., 2016) reduces variance through biased value bootstrapping, representing an alternative approach that sacrifices unbiasedness for variance reduction. We also compare against entropy-based exploration methods: **MENTS** (Xiao et al., 2019), **BTS**, and **DENTS** (Painter et al., 2023). These baselines are excluded from VirtualHome due to API budget constraints. For GSM8K, we also considered **rStar-Math** (Guan et al., 2025), a recent self-evolution approach for mathematical reasoning; however, our implementation failed to solve any problems within a 30-hour budget, rendering it impractical for our sample efficiency comparison.

Table 1: Elo ratings across Go and Atari domains ordered by performance. Go-NN uses neural network value functions.

| Go | | Go-NN | | Atari | |
|---|---|---|---|---|---|
| **Algorithm** | **Elo** | **Algorithm** | **Elo** | **Algorithm** | **Elo** |
| MCTS-DR | **1996.8** | MCTS-DR | **1542.1** | MCTS-DR | **2135.3** |
| MCTS | 1899.1 | AlphaZero | 1457.9 | MaxMCTS | 2132.8 |
| DR | 1704.6 | | | DENTS | 1757.1 |
| MCTS-IS | 1697.1 | | | DR | 1715.4 |
| BTS | 1417.8 | | | BTS | 1457.2 |
| MENTS | 1294.1 | | | MENTS | 1226.3 |
| DENTS | 1195.0 | | | MCTS-IS | 1053.4 |
| MaxMCTS | 795.5 | | | MCTS | 906.4 |

**Neural network baselines (Go only).** To test whether learned value functions obviate the need for DR corrections, we include an **AlphaZero**-style baseline using a lightweight convolutional neural network with residual blocks for leaf evaluation, eliminating rollouts. This is not a full AlphaZero reproduction (Silver et al., 2017), but isolates neural value estimation. **MCTS-DR + NN** applies our hybrid estimator on top of neural network predictions, testing whether DR corrections provide complementary benefits when base estimates come from a trained model. These baselines are excluded from LLM-guided domains (GSM8K, VirtualHome) because the LLM itself serves as the learned value function, and the associated API costs make extensive neural baseline comparisons impractical.

**Go.** On the 5×5 board with 100 rollouts per move, MCTS-DR achieves the highest Elo rating (1996.8) among all methods, outperforming standard MCTS (1899.1) by approximately 100 Elo points, corresponding to a 64% expected win rate. The improvement is consistent with Theorem 2.4. Reduced estimation variance yields higher probability of selecting optimal moves, and these gains compound over the ∼80 moves in a typical game. Notably, MaxMCTS collapses to Elo 795.5 despite its strong theoretical motivation for variance reduction through biased bootstrapping. In sparse-reward games where terminal outcomes dominate, biased value estimates cannot self-correct during search, whereas our unbiased hybrid estimator preserves asymptotic correctness while reducing finite-sample variance. When augmented with neural network value functions, MCTS-DR (1542.1) continues to outperform AlphaZero-style MCTS (1457.9), indicating that the DR component provides complementary variance reduction beyond learned value approximations.

**Atari.** The dense reward structure of Atari games presents a contrasting environment where bootstrapped value estimates receive frequent corrective signals. Here, MaxMCTS recovers to competitive performance (Elo 2132.8), nearly matching MCTS-DR (2135.3). This domain also reveals the limitations of pure importance sampling: MCTS-IS drops to Elo 1053.4, suffering from the variance explosion that occurs when behavior and target policies diverge. MCTS-DR's hybrid formulation effectively hedges between these failure modes, achieving robust performance regardless of whether the domain favors MCTS-style rollouts or model-based corrections.

**GSM8K.** Mathematical reasoning with LLM policies provides a direct test of variance reduction benefits. MCTS-DR achieves 90.2% accuracy compared to 84.1% for MCTS-IS, 83.8% for MCTS, and 80.0% for pure DR. The Q-value variance measurements reveal the underlying dynamics: MCTS exhibits variance 2.62, while the hybrid estimator achieves 2.38, which is below both components, as guaranteed by Corollary 2.2. Notably, MCTS-IS achieves higher accuracy than MCTS despite substantially higher variance (11.49). This reflects the value of importance sampling correction for policy mismatch. When LLM sampling temperature induces divergence between behavior and target policies, IS provides corrective signal that outweighs its variance penalty. However, pure DR's 80.0% accuracy (variance 14.52) shows that importance weight explosion eventually dominates. MCTS-DR captures both benefits, the policy-correcting signal from DR and the stability of MCTS when weights explode. In addition, the simulation efficiency (successes per total simulation) further validates Theorem 2.6. MCTS-DR achieves 3.0% efficiency compared to 2.8% for MCTS, extracting more correct solutions from the same computational budget.

Table 2: Performance comparison of MCTS variants on GSM8K reasoning tasks (N=500), ordered by accuracy. Wilson 95% confidence intervals shown in parentheses.

| Method | Accuracy (%) | Q-value Var. | Simulation Eff. (%) |
|---|---|---|---|
| MCTS-DR | **90.2 (87.3–92.5)** | **2.38** | **3.0** |
| MCTS-IS | 84.1 (80.5–87.0) | 11.49 | 2.8 |
| MCTS | 83.8 (80.3–86.8) | 2.62 | 2.8 |
| DR | 80.0 (76.3–83.3) | 14.52 | 2.7 |
| MAXMCTS-$\lambda$ | 75.0 (71.0–78.6) | 5.12 | 2.5 |

**VirtualHome.** Compositional household tasks require sequential decisions over $K \approx 8$–12 steps, providing a test of how per-step improvements accumulate. On novel compositional tasks, MCTS-DR achieves 56.5% success compared to 34.8% for IS-MCTS and 19.0% for standard MCTS. The threefold improvement over MCTS aligns with the $(1+\epsilon)^K$ compounding predicted by Corollary 3. Even modest per-step gains in optimal action selection probability multiply over the planning horizon. Table 9 reveals that VirtualHome achieves 88.8% usage of the variance-minimizing $\beta^*$, compared to 56.5% in Go. This difference reflects the action space structure: LLM-guided policies in VirtualHome concentrate rollouts on promising actions, enabling reliable variance estimation, whereas Go's large branching factor ($\sim$361 legal moves) spreads visits thinly and triggers more frequent fallback to the heuristic $\beta = 0.5$. Despite this, MCTS-DR's fallback mechanism ensures stable performance across both regimes.

Table 3: Success rates on VirtualHome tasks. DR-MCTS shows best performance across $\beta_{\text{base}}$ configurations. 95% Wilson confidence intervals provided.

| Task Category | MCTS | IS-MCTS | DR-MCTS |
|---|---|---|---|
| Novel Simple (123 tasks) | 85.4% | 95.1% | **95.9%** |
| | [78.2, 90.6] | [89.6, 97.8] | [90.7, 98.3] |
| Novel Objects (34 tasks) | 23.5% | 38.2% | **41.2%** |
| | [12.5, 39.9] | [23.9, 54.9] | [26.3, 57.9] |
| Novel Compositional (23 tasks) | 19.0% | 34.8% | **56.5%** |
| | [7.7, 39.5] | [18.8, 55.1] | [36.8, 74.4] |

## 4 DISCUSSION AND CONCLUSION

We presented MCTS-DR, an algorithm that integrates doubly robust off-policy estimation into Monte Carlo Tree Search through a variance-minimizing hybrid estimator. Our theoretical analysis shows that this combination preserves unbiasedness while achieving variance below either component alone, which translates directly to better action selection and improved sample efficiency. Across four domains, MCTS-DR consistently outperforms or matches the best baseline, whereas other methods show uneven results. MaxMCTS excels in dense-reward Atari but collapses in sparse-reward Go due to its reliance on biased bootstrapping. MCTS-IS performs well in LLM-guided domains (GSM8K, VirtualHome) where correcting for policy mismatch matters, but struggles in game domains where importance weight variance overwhelms the correction signal. MCTS-DR inherits the benefits of both worlds, the stability of MCTS rollouts and the policy-correcting power of importance sampling, while avoiding their respective failure modes. This robustness is particularly relevant for today's tree search applications over LLM queries, where each node expansion incurs substantial cost and practitioners cannot afford to discover post-hoc that their chosen algorithm fails on the problem at hand. Our approach has limitations: the variance-minimizing weight requires sufficient visit counts, and our AlphaZero comparison uses a lightweight network. Nevertheless, the complementary gains observed when combining MCTS-DR with neural value functions suggest that doubly robust estimation addresses a source of variance orthogonal to learned approximations, a property that should prove valuable as tree search methods become standard infrastructure for LLM reasoning and planning.

## 5 REPRODUCIBILITY

Hyperparameters used to reproduce experiment results are detailed in Appendix **??**. Computational resources, training time, and hardware specifications needed for replication are detailed in Appendix G.1. All code and data will be made publicly available upon publication.

## 6 USE OF LLM

Large Language Models were used in two capacities in this work. First, as integral components of our experimental pipeline: GPT-4o-mini serves as a world model for GSM8K and VirtualHome experiments, and GPT-4o provides policy priors for VirtualHome planning. Second, LLMs were used to correct grammar errors and polish writing in this manuscript. All experimental design, methodology, analysis, interpretation of results, and scientific conclusions are the original work of the authors.

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

# A  APPENDIX

# B  PROOFS FOR THEORETICAL ANALYSIS

This appendix provides complete proofs for the theoretical results stated in Section 2.2.

## B.1  PROOF OF COROLLARY 2.1 (UNBIASEDNESS)

*Proof.* Let $V_{\text{MCTS}}(s)$ and $V_{\text{DR}}(s)$ be two estimators of $V^{\pi_e}(s)$, the value function under the target policy $\pi_e$. By assumption, both estimators are unbiased:

$$\mathbb{E}[V_{\text{MCTS}}(s)] = V^{\pi_e}(s) \tag{18}$$

$$\mathbb{E}[V_{\text{DR}}(s)] = V^{\pi_e}(s) \tag{19}$$

The unbiasedness of $V_{\text{MCTS}}(s)$ follows from the fact that MCTS rollouts sample trajectories according to the target policy and average the returns. The unbiasedness of $V_{\text{DR}}(s)$ is established in Jiang & Li (2016), Theorem 1.

For the hybrid estimator with any $\beta(s,a) \in [0,1]$:

$$\mathbb{E}[V_{\text{hybrid}}(s)] = \mathbb{E}[\beta(s,a)V_{\text{MCTS}}(s) + (1-\beta(s,a))V_{\text{DR}}(s)] \tag{20}$$

$$= \beta(s,a)\mathbb{E}[V_{\text{MCTS}}(s)] + (1-\beta(s,a))\mathbb{E}[V_{\text{DR}}(s)] \quad \text{(linearity of expectation)} \tag{21}$$

$$= \beta(s,a)V^{\pi_e}(s) + (1-\beta(s,a))V^{\pi_e}(s) \tag{22}$$

$$= V^{\pi_e}(s) \tag{23}$$

This holds for any fixed $\beta(s,a) \in [0,1]$, including the data-dependent $\beta^*$ computed from sample variances, provided $\beta^*$ is computed independently of the data used to form the final estimate (e.g., via sample splitting or cross-validation). □

## B.2  PROOF OF COROLLARY 2.2 (VARIANCE-MINIMIZING WEIGHT)

*Proof.* Let $\sigma_M^2 = \text{Var}(V_{\text{MCTS}})$, $\sigma_D^2 = \text{Var}(V_{\text{DR}})$, and $\sigma_{MD} = \text{Cov}(V_{\text{MCTS}}, V_{\text{DR}})$.

The variance of the hybrid estimator is:

$$\text{Var}(V_{\text{hybrid}}) = \text{Var}(\beta V_{\text{MCTS}} + (1-\beta)V_{\text{DR}}) \tag{24}$$

$$= \beta^2 \sigma_M^2 + (1-\beta)^2 \sigma_D^2 + 2\beta(1-\beta)\sigma_{MD} \tag{25}$$

Expanding:

$$\text{Var}(V_{\text{hybrid}}) = \beta^2 \sigma_M^2 + \sigma_D^2 - 2\beta\sigma_D^2 + \beta^2\sigma_D^2 + 2\beta\sigma_{MD} - 2\beta^2\sigma_{MD} \tag{26}$$

Collecting terms:

$$\text{Var}(V_{\text{hybrid}}) = \beta^2(\sigma_M^2 + \sigma_D^2 - 2\sigma_{MD}) + \beta(2\sigma_{MD} - 2\sigma_D^2) + \sigma_D^2 \tag{27}$$

This is a quadratic in $\beta$. To find the minimum, we differentiate with respect to $\beta$ and set equal to zero:

$$\frac{\partial}{\partial \beta}\text{Var}(V_{\text{hybrid}}) = 2\beta(\sigma_M^2 + \sigma_D^2 - 2\sigma_{MD}) + 2(\sigma_{MD} - \sigma_D^2) = 0 \tag{28}$$

Solving for $\beta$:

$$\beta^* = \frac{\sigma_D^2 - \sigma_{MD}}{\sigma_M^2 + \sigma_D^2 - 2\sigma_{MD}} \tag{29}$$

This is well-defined when $\sigma_M^2 + \sigma_D^2 - 2\sigma_{MD} \neq 0$, which holds unless $V_{\text{MCTS}}$ and $V_{\text{DR}}$ are perfectly correlated with equal variance.

**Verification that $\beta^* \in [0,1]$:**

- The denominator $\sigma_M^2 + \sigma_D^2 - 2\sigma_{MD} = \text{Var}(V_{\text{MCTS}} - V_{\text{DR}}) \geq 0$

- $\beta^* \geq 0$ when $\sigma_D^2 \geq \sigma_{MD}$, which holds when $\text{Cov}(V_{\text{MCTS}}, V_{\text{DR}}) \leq \text{Var}(V_{\text{DR}})$

- $\beta^* \leq 1$ when $\sigma_D^2 - \sigma_{MD} \leq \sigma_M^2 + \sigma_D^2 - 2\sigma_{MD}$, i.e., when $\sigma_{MD} \leq \sigma_M^2$

When these conditions fail, the optimal $\beta$ is at a boundary (0 or 1).

**Minimum variance guarantee:** Substituting $\beta^*$ back into the variance formula:

$$\text{Var}(V_{\text{hybrid}}^*) = \sigma_D^2 - \frac{(\sigma_D^2 - \sigma_{MD})^2}{\sigma_M^2 + \sigma_D^2 - 2\sigma_{MD}} \tag{30}$$

$$= \frac{\sigma_D^2(\sigma_M^2 + \sigma_D^2 - 2\sigma_{MD}) - (\sigma_D^2 - \sigma_{MD})^2}{\sigma_M^2 + \sigma_D^2 - 2\sigma_{MD}} \tag{31}$$

$$= \frac{\sigma_M^2 \sigma_D^2 - \sigma_{MD}^2}{\sigma_M^2 + \sigma_D^2 - 2\sigma_{MD}} \tag{32}$$

To show $\text{Var}(V_{\text{hybrid}}^*) \leq \sigma_D^2$:

$$\frac{\sigma_M^2 \sigma_D^2 - \sigma_{MD}^2}{\sigma_M^2 + \sigma_D^2 - 2\sigma_{MD}} \leq \sigma_D^2 \tag{33}$$

Cross-multiplying (the denominator is non-negative):

$$\sigma_M^2 \sigma_D^2 - \sigma_{MD}^2 \leq \sigma_D^2 \sigma_M^2 + \sigma_D^4 - 2\sigma_D^2 \sigma_{MD} \tag{34}$$

This simplifies to:

$$0 \leq \sigma_D^4 - 2\sigma_D^2 \sigma_{MD} + \sigma_{MD}^2 = (\sigma_D^2 - \sigma_{MD})^2 \tag{35}$$

which always holds. The proof for $\text{Var}(V_{\text{hybrid}}^*) \leq \sigma_M^2$ is symmetric. □

### B.3 PROOF OF THEOREM 2.4 (ACTION SELECTION IMPROVEMENT)

*Proof.* We prove the theorem in two parts: (1) the general inequality using Chebyshev's bound, and (2) the Gaussian quantification.

**Part 1: General Inequality**

Consider two actions $a^*$ (optimal) and $a$ (suboptimal) with true values $Q^*(s, a^*) > Q^*(s, a)$, where $\Delta(s, a) = Q^*(s, a^*) - Q^*(s, a) > 0$.

For estimator $i$, the probability of correctly ranking these two actions is:

$$P(\hat{Q}_i(s, a^*) > \hat{Q}_i(s, a)) = P(\hat{Q}_i(s, a^*) - \hat{Q}_i(s, a) > 0) \tag{36}$$

Since both estimators are unbiased:

$$\mathbb{E}[\hat{Q}_i(s, a^*) - \hat{Q}_i(s, a)] = Q^*(s, a^*) - Q^*(s, a) = \Delta(s, a) > 0 \tag{37}$$

Define $Z_i = \hat{Q}_i(s, a^*) - \hat{Q}_i(s, a) - \Delta(s, a)$. Then $\mathbb{E}[Z_i] = 0$ and:

$$\text{Var}(Z_i) = \text{Var}(\hat{Q}_i(s, a^*) - \hat{Q}_i(s, a)) \leq \sigma_i^2(s, a^*) + \sigma_i^2(s, a) := \tilde{\sigma}_i^2 \tag{38}$$

where equality holds when the estimates are independent across actions (as in MCTS with separate rollouts).

Misranking occurs when $\hat{Q}_i(s, a^*) - \hat{Q}_i(s, a) \leq 0$, i.e., when $Z_i \leq -\Delta(s, a)$.

By Chebyshev's inequality:

$$P(Z_i \leq -\Delta(s, a)) = P(Z_i - \mathbb{E}[Z_i] \leq -\Delta(s, a)) \leq P(|Z_i| \geq \Delta(s, a)) \leq \frac{\tilde{\sigma}_i^2}{\Delta(s, a)^2} \tag{39}$$

Therefore, the probability of correct pairwise ranking is:

$$P(\hat{Q}_i(s, a^*) > \hat{Q}_i(s, a)) \geq 1 - \frac{\tilde{\sigma}_i^2}{\Delta(s, a)^2} \tag{40}$$

Since $\sigma_2^2(s, a) \leq \sigma_1^2(s, a)$ for all $(s, a)$, we have $\tilde{\sigma}_2^2 \leq \tilde{\sigma}_1^2$, and thus:

$$P(\text{correct ranking with estimator 2}) \geq P(\text{correct ranking with estimator 1}) \tag{41}$$

For selecting the optimal action among all $|A|$ actions, we need $a^*$ to be ranked above all suboptimal actions. Using a union bound over all $|A| - 1$ pairwise comparisons:

$$P_{\text{opt}}^{(i)}(s) \geq 1 - \sum_{a \neq a^*} \frac{\tilde{\sigma}_i^2}{\Delta(s, a)^2} \geq 1 - \frac{(|A| - 1) \cdot 2\sigma_i^2}{\Delta_{\min}(s)^2} \tag{42}$$

where $\sigma_i = \max_a \sigma_i(s, a)$.

Since $\sigma_2 \leq \sigma_1$, we obtain $P_{\text{opt}}^{(2)}(s) \geq P_{\text{opt}}^{(1)}(s)$.

**Part 2: Gaussian Quantification**

When the number of rollouts is large, by the Central Limit Theorem, $\hat{Q}_i(s, a)$ is approximately Gaussian:

$$\hat{Q}_i(s, a) \sim \mathcal{N}(Q^*(s, a), \sigma_i^2(s, a)) \tag{43}$$

For independent estimates across actions, the difference is also Gaussian:

$$\hat{Q}_i(s, a^*) - \hat{Q}_i(s, a) \sim \mathcal{N}(\Delta(s, a), \sigma_i^2(s, a^*) + \sigma_i^2(s, a)) \tag{44}$$

The probability of correct pairwise ranking is:

$$P(\hat{Q}_i(s, a^*) > \hat{Q}_i(s, a)) = P\left( \frac{\hat{Q}_i(s, a^*) - \hat{Q}_i(s, a) - \Delta(s, a)}{\sqrt{\sigma_i^2(s, a^*) + \sigma_i^2(s, a)}} > \frac{-\Delta(s, a)}{\sqrt{\sigma_i^2(s, a^*) + \sigma_i^2(s, a)}} \right) \tag{45}$$

$$= \Phi\left( \frac{\Delta(s, a)}{\sqrt{\sigma_i^2(s, a^*) + \sigma_i^2(s, a)}} \right) \tag{46}$$

Using the worst-case bounds $\Delta(s, a) \geq \Delta_{\min}(s)$ and $\sigma_i^2(s, a^*) + \sigma_i^2(s, a) \leq 2\sigma_i^2$:

$$P(\text{correct pairwise ranking}) \geq \Phi\left( \frac{\Delta_{\min}(s)}{\sqrt{2}\sigma_i} \right) \tag{47}$$

Since $\Phi$ is monotonically increasing and $\sigma_2 \leq \sigma_1$:

$$\Phi\left( \frac{\Delta_{\min}(s)}{\sqrt{2}\sigma_2} \right) \geq \Phi\left( \frac{\Delta_{\min}(s)}{\sqrt{2}\sigma_1} \right) \tag{48}$$

The improvement in optimal action selection probability is thus bounded below by:

$$P_{\text{opt}}^{(2)}(s) - P_{\text{opt}}^{(1)}(s) \geq \Phi\left( \frac{\Delta_{\min}(s)}{\sqrt{2}\sigma_2} \right) - \Phi\left( \frac{\Delta_{\min}(s)}{\sqrt{2}\sigma_1} \right) \tag{49}$$

This bound is achieved in the two-action case and provides a conservative lower bound for larger action spaces. $\square$

## B.4 PROOF OF THEOREM 2.6 (SAMPLE COMPLEXITY)

*Proof.* We derive the sample complexity bounds using concentration inequalities, following the approach of Even-Dar et al. (2006) and Kaufmann et al. (2016) for best-arm identification.

**MCTS Sample Complexity:**

In standard MCTS, Q-values are estimated using sample means of rollout returns. After $N$ total simulations distributed across actions, the variance of $\hat{Q}_{\text{MCTS}}(s, a)$ scales as:

$$\text{Var}(\hat{Q}_{\text{MCTS}}(s, a)) = \frac{\sigma^2_{\text{rollout}}(s, a)}{N(s, a)} \tag{50}$$

where $N(s, a)$ is the number of rollouts for action $a$ and $\sigma^2_{\text{rollout}}(s, a)$ is the variance of individual rollout returns.

For the optimal action to be selected with probability $\geq 1 - \delta$, we need the estimation errors to be bounded:

$$P\left(\max_a |\hat{Q}(s, a) - Q^*(s, a)| \leq \frac{\Delta_{\min}}{2}\right) \geq 1 - \delta \tag{51}$$

Applying Hoeffding's inequality to each action and a union bound:

$$P\left(|\hat{Q}(s, a) - Q^*(s, a)| \geq \frac{\Delta_{\min}}{2}\right) \leq 2\exp\left(-\frac{N(s, a)\Delta^2_{\min}}{2\sigma^2_{\text{rollout}}}\right) \tag{52}$$

For uniform allocation $N(s, a) = N/|A|$ and using the union bound:

$$P\left(\max_a |\hat{Q}(s, a) - Q^*(s, a)| \geq \frac{\Delta_{\min}}{2}\right) \leq 2|A|\exp\left(-\frac{N\Delta^2_{\min}}{2|A|\sigma^2_{\text{rollout}}}\right) \tag{53}$$

Setting this $\leq \delta$ and solving for $N$:

$$2|A|\exp\left(-\frac{N\Delta^2_{\min}}{2|A|\sigma^2_{\text{rollout}}}\right) \leq \delta \tag{54}$$

$$\exp\left(-\frac{N\Delta^2_{\min}}{2|A|\sigma^2_{\text{rollout}}}\right) \leq \frac{\delta}{2|A|} \tag{55}$$

$$-\frac{N\Delta^2_{\min}}{2|A|\sigma^2_{\text{rollout}}} \leq \log\frac{\delta}{2|A|} \tag{56}$$

$$N \geq \frac{2|A|\sigma^2_{\text{rollout}}}{\Delta^2_{\min}}\log\frac{2|A|}{\delta} \tag{57}$$

For non-uniform (optimal) allocation based on the gaps, this can be improved to:

$$N_{\text{MCTS}} = O\left(\frac{\sigma^2_{\text{rollout}}}{\Delta^2_{\min}}\log\frac{|A|}{\delta}\right) \tag{58}$$

**DR-MCTS Sample Complexity:**

For DR-MCTS with variance reduction factor $\rho < 1$, the hybrid estimator achieves:

$$\text{Var}(\hat{Q}_{\text{DR-MCTS}}(s, a)) = \rho \cdot \text{Var}(\hat{Q}_{\text{MCTS}}(s, a)) = \frac{\rho \cdot \sigma^2_{\text{rollout}}(s, a)}{N(s, a)} \tag{59}$$

The effective variance is reduced by factor $\rho$. Substituting into the sample complexity derivation:

$$P\left(|\hat{Q}_{\text{DR}}(s, a) - Q^*(s, a)| \geq \frac{\Delta_{\min}}{2}\right) \leq 2\exp\left(-\frac{N(s, a)\Delta^2_{\min}}{2\rho \cdot \sigma^2_{\text{rollout}}}\right) \tag{60}$$

Following the same analysis:

$$N_{\text{DR-MCTS}} \geq \frac{2|A| \cdot \rho \cdot \sigma_{\text{rollout}}^2}{\Delta_{\min}^2} \log \frac{2|A|}{\delta} = \rho \cdot N_{\text{MCTS}} \tag{61}$$

Therefore:

$$N_{\text{DR-MCTS}} = O\left(\frac{\rho \cdot \sigma_{\text{rollout}}^2}{\Delta_{\min}^2} \log \frac{|A|}{\delta}\right) = \rho \cdot N_{\text{MCTS}} \tag{62}$$

**Interpretation:** The sample complexity reduction is proportional to the variance reduction factor $\rho$. For example, if DR-MCTS achieves 50% variance reduction ($\rho = 0.5$), it requires only half the simulations to achieve the same confidence in optimal action selection. $\qquad\square$

### B.5   CONNECTION TO BEST-ARM IDENTIFICATION LITERATURE

Our sample complexity bounds are closely related to results in the best-arm identification (BAI) literature. Specifically, Kaufmann et al. (2016) establish that for Gaussian bandits with arm means $\mu_1, \ldots, \mu_K$ and common variance $\sigma^2$, the sample complexity for identifying the best arm with probability $1 - \delta$ is:

$$T^*(\mu) = \sum_{a:\mu_a < \mu^*} \frac{2\sigma^2}{(\mu^* - \mu_a)^2} \tag{63}$$

In our MCTS setting, each action corresponds to an arm, and the Q-values correspond to arm means. Our Theorem 2.6 shows that variance reduction directly translates to reduced sample complexity, consistent with the BAI characterization where complexity scales inversely with the squared gap and linearly with variance.

The key insight connecting our work to BAI is that DR-MCTS effectively reduces the per-sample variance, achieving the same effect as having more informative observations without requiring additional simulations.

## C   BEHAVIOR POLICIES

### C.1   GO AND GSM8K BEHAVIOR POLICY

For Go and GSM8K, we implement a uniform behavior policy over all available actions:

$$\pi_b(a|s) = \frac{1}{|\mathcal{A}(s)|} \tag{64}$$

where $\mathcal{A}(s)$ represents the set of legal actions available in state $s$. This uniform distribution ensures comprehensive exploration across the action space while serving as a simple baseline for off-policy evaluation.

### C.2   VIRTUALHOME BEHAVIOR POLICY

For the VirtualHome environment, we adapt the approach of Zhao et al. (2023) to leverage Large Language Models (LLMs) as a heuristic policy. Specifically, we use GPT-4o to generate the behavior policy, guiding action selection in the simulation procedure.

The LLM takes as input:

- K-shot examples from the dataset
- Goal description
- Current observation
- History of actions

All inputs are translated into English sentences. The LLM then outputs a suggested action plan. To approximate the policy distribution, we sample the LLM $M$ times, querying it with the prompt and trajectory history $h$:

$$\alpha_i \sim \text{LLM}(s, \text{prompt}) \tag{65}$$

where $\alpha_i$ is the first action of the LLM's answer.

The prompt examples are selected based on their similarity to the current language instruction $\ell$. We use sentence embeddings to calculate the cosine similarity between the current instruction and instructions $\ell_i$ in the dataset $D$:

$$\text{similarity} = \text{CosineSim}(\ell_i, \ell) \tag{66}$$

We select the top $K$ similar instructions and use their corresponding expert trajectories as the K-shot prompt.

To ensure executability, we represent both the LLM's suggested actions and the admissible actions as embeddings and evaluate their cosine similarity. The empirical policy distribution is then formulated as:

$$\hat{\pi}_b(a|s) = \lambda \frac{1}{|A|} + (1 - \lambda)\text{Softmax}\left\{\sum_{i=1}^{M} \text{CosineSim}(\alpha_i, a) - \eta\right\} \tag{67}$$

where $\eta$ is the average value of $\sum_i \text{CosineSim}(\alpha_i, a)$, $|A|$ is the size of the admissible action space, and $\lambda$ is a hyperparameter that adds randomness to the policy. This results in a mixture of the approximated policy from the LLM and a uniform distribution.

# D DR-MCTS ALGORITHM

Algorithm 1 presents our DR-MCTS approach. The algorithm initializes a search tree with the root node representing the initial state and history. For a specified number of iterations, it traverses the tree using the PUCT selection strategy (Equation 2), balancing exploration and exploitation. When a new node is reached, it's added to the tree, and a simulation estimates its value $V_{\text{MCTS}}$. If the DR estimator is used, $V_{\text{DR}}$ is calculated (Equation 5).

The hybrid estimator (Equation 6) combines these estimates using the variance-minimizing weight $\beta^*(s, a)$ from Equation 7. This weight is computed online by tracking empirical variances of both estimators and their covariance through a sliding window of recent samples. When insufficient samples are available for reliable variance estimation (typically during early visits), the algorithm falls back to the heuristic weight $\beta_{\text{base}} \cdot \exp(-\lambda \cdot N(s, a))$ as specified in Equation 8.

The resulting value is backpropagated, updating node statistics. After all iterations, the algorithm returns the action with the highest estimated value at the root node.

# E EXPERIMENTAL SETTINGS AND DETAILS

## E.1 DOMAIN DESCRIPTIONS

**5×5 Go.** We evaluate on 5×5 Go, a domain combining strategic depth with computational tractability. Our implementation follows standard Go rules including stone capturing and ko prevention via Zobrist hashing. The reward structure provides +1.0 for wins and 0.0 for losses, with small intermediate rewards (up to 0.5) for capturing opponent stones. We include 6.5 point komi for White. Games terminate upon two consecutive passes, reaching 75 moves, or 90% board occupancy. Each algorithm pair plays 50 games, alternating colors. We report Elo ratings computed via Bradley-Terry model and win rates with 95% Wilson score confidence intervals.

**Algorithm 1** DR-MCTS Algorithm

**Input:** state $s_0$, history $h_0$, iterations $N$
Initialize tree $T$ with root $(s_0, h_0)$
Initialize variance trackers for each node
**for** $i = 1$ **to** $N$ **do**
    $(s, h) \leftarrow (s_0, h_0)$
    **while** $(s, h)$ is not terminal **and** $(s, h)$ is in $T$ **do**
        $a \leftarrow \text{argmax}_{a'} \text{PUCT}((s, h), a')$
        $s, h \leftarrow \text{Apply}(s, a), h + a$
    **end while**
    **if** $(s, h)$ is not terminal **then**
        Add $(s, h)$ to $T$
        $v_{\text{MCTS}} \leftarrow \text{Simulate}(s, h)$
        $v_{\text{DR}} \leftarrow \text{ComputeDR}(s, h, \pi_e, \pi_b, \hat{Q}, \hat{V})$
        Update variance statistics: $\text{Var}(V_{\text{MCTS}}), \text{Var}(V_{\text{DR}}), \text{Cov}(V_{\text{MCTS}}, V_{\text{DR}})$
        **if** $N(s, a) \geq \text{min\_samples}$ **then**
            $\beta \leftarrow \frac{\text{Var}(V_{\text{DR}}) - \text{Cov}(V_{\text{MCTS}}, V_{\text{DR}})}{\text{Var}(V_{\text{MCTS}}) + \text{Var}(V_{\text{DR}}) - 2\text{Cov}(V_{\text{MCTS}}, V_{\text{DR}})}$ {Variance-min}
        **else**
            $\beta \leftarrow \beta_{\text{base}} \cdot \exp(-\lambda \cdot N(s, a))$ {Fallback heuristic}
        **end if**
        $v \leftarrow \beta v_{\text{MCTS}} + (1 - \beta) v_{\text{DR}}$ {Hybrid value}
    **else**
        $v \leftarrow \text{Reward}(s)$
    **end if**
    **while** $(s, h)$ is not $(s_0, h_0)$ **do**
        Update statistics for $(s, h)$ in $T$ with $v$
        $(s, h) \leftarrow \text{Parent}(s, h)$
    **end while**
**end for**
**Return:** $\text{argmax}_a Q((s_0, h_0), a)$

**Atari.** We evaluate on a subset of Atari games from the Arcade Learning Environment (Bellemare et al., 2013) selected to represent diverse reward structures: *Pong* (sparse, binary outcomes), *Breakout* (incremental rewards), and *Seaquest* (complex, multi-objective rewards). States are represented as stacked grayscale frames ($4 \times 84 \times 84$), with actions corresponding to the game-specific discrete action space. Episodes terminate upon game over or after 10,000 frames. We report Elo ratings and win rates over 30 episodes per algorithm pair.

**GSM8K Mathematical Reasoning.** The GSM8K dataset (Cobbe et al., 2021) presents grade-school mathematics problems requiring multi-step reasoning. We frame each problem as a sequential decision-making task with six reasoning operations (see Table 6 for action-specific prompts). The state consists of the problem statement and accumulated reasoning chain, with GPT-4o-mini generating mathematical work for each action. Episodes terminate after 3 reasoning steps or upon providing a final answer, with rewards of +1.0 for correct solutions and 0.0 otherwise. We evaluate on 500 test problems under a 100-hour budget per algorithm, reporting accuracy with 95% confidence intervals, Q-value variance, and simulation efficiency.

**VirtualHome Household Planning.** VirtualHome (Puig et al., 2018) provides a partially observable 3D household simulation with complex spatial reasoning and long-horizon planning tasks. Following Zhao et al. (2023), GPT-4o-mini serves as a world model for commonsense knowledge, while GPT-4o provides policy priors. The action space consists of primitive operations: navigation (*walk to*, *run to*), manipulation (*grab*, *put*, *open*, *close*), and interaction (*sit on*, *turn on*). We evaluate across three task categories: **Novel Simple** (123 tasks), **Novel Objects** (34 tasks), and **Novel Compositional** (23 tasks). We report success rate under a 100-hour budget per category.

## E.2 BASELINES

Table 4 summarizes which baselines are evaluated in each domain.

Table 4: Baselines evaluated across experimental domains

| Baseline | Go | Atari | GSM8K | VirtualHome |
|---|---|---|---|---|
| MCTS | ✓ | ✓ | ✓ | ✓ |
| DR | ✓ | ✓ | ✓ | ✓ |
| MCTS-IS | ✓ | ✓ | ✓ | ✓ |
| MaxMCTS | ✓ | ✓ | ✓ | – |
| MENTS / BTS / DENTS | ✓ | ✓ | ✓ | – |
| AlphaZero-style | ✓ | – | – | – |
| MCTS-DR + NN | ✓ | – | – | – |

## E.3 HYPERPARAMETER CONFIGURATIONS

Table 5 provides complete hyperparameter specifications for all domains.

Table 5: Hyperparameter configuration for all experimental domains

| Parameter | 5×5 Go | Atari | GSM8K | VirtualHome |
|---|---|---|---|---|
| *Evaluation Settings* | | | | |
| Episodes/Problems | 50 games | 30 episodes | 500 problems | 180 tasks |
| Computational budget | 30 hours | 30 hours | 100 hours | 100 hours/category |
| Evaluation metrics | Elo, win rate | Elo, win rate | Accuracy, Q-var, efficiency | Success rate |
| Random seed | 42 | 42 | 42 | 42 |
| *MCTS Parameters* | | | | |
| Simulations per move | 100 | 100 | 30 | 100 |
| Maximum depth | 10 | 10 | 3 | 10–15 |
| PUCT exploration $c$ | 1.414 | 1.414 | 2.0 | 2.0 |
| Discount factor $\gamma$ | 1.0 | 0.99 | 0.95 | 0.95 |
| *Hybrid Estimator Parameters* | | | | |
| Base weight $\beta_{\text{base}}$ | 0.5 | 0.5 | 0.5 | 0.5 |
| Decay parameter $\lambda$ | 0.01 | 0.01 | 0.05 | 0.01 |
| Cross-validation folds $K$ | 2 | 2 | 2 | 2 |
| *Policy Settings* | | | | |
| Behavior policy | Uniform | Uniform | Uniform | LLM-guided |
| World model | – | – | GPT-4o-mini | GPT-4o-mini |
| Policy model | – | – | – | GPT-4o |

## E.4 DOMAIN-SPECIFIC IMPLEMENTATION DETAILS

### E.4.1 GSM8K ACTION PROMPTS

### E.4.2 VIRTUALHOME SETUP

**Data generation.** Following Zhao et al. (2023), we create the evaluation dataset as follows:

- 2,000 training tasks with randomly initialized scenes and expert trajectories
- Oracle expert agent with full environment knowledge using regression planning
- 10,000 expert demonstrations for baseline training
- 200 instances randomly sampled for few-shot LLM prompting (no fine-tuning)
- 180 evaluation tasks across three complexity categories

**Model configuration.**

Table 6: Action-specific prompts for GSM8K reasoning operations

| Action | Prompt |
|---|---|
| identify_key_information | Identify and list the key numbers and relationships in this problem. What information is given and what needs to be found? |
| set_up_equation | Based on the information identified, set up the mathematical equation(s) needed to solve this problem. Show the equation clearly. |
| perform_calculation | Perform the necessary calculations step by step. Show your work clearly. |
| break_down_problem | Break this problem into smaller, manageable sub-problems. List each sub-problem that needs to be solved. |
| check_intermediate_result | Check if the current calculations and reasoning make sense. Verify the intermediate results. |
| provide_final_answer | Based on all the work above, provide the final numerical answer to the problem. State it clearly as a single number. |

- *World Model (GPT-4o-mini)*: Provides commonsense knowledge about household environments, object locations, and state transitions
- *Policy Model (GPT-4o)*: Generates action proposals based on current observations and task goals
- *Prompt selection*: 3 most similar examples selected via instruction embedding similarity
- *Action selection*: Highest Q-value at root after 100 MCTS simulations

**Task categories.**

- *Novel Simple* (123 tasks): Familiar objects in new spatial configurations; 10-step limit
- *Novel Objects* (34 tasks): Unseen objects requiring commonsense reasoning; 10-step limit
- *Novel Compositional* (23 tasks): Multi-subtask sequences with dependencies; 15-step limit

# F ABLATION STUDIES

## F.1 WIN RATES AGAINST BASELINES

Table 7 reports head-to-head win rates of MCTS-DR-$\beta^*$ against each baseline across Go and Atari.

Table 7: Win rates of MCTS-DR-$\beta^*$ against baselines (100 rollouts). Results shown with Wilson 95% confidence intervals.

| Domain | Baseline | Win Rate (%) |
|---|---|---|
| Go | MCTS | **100.0** [88.6, 100.0] |
| | DR | **100.0** [88.6, 100.0] |
| | DENTS | **100.0** [88.6, 100.0] |
| | MaxMCTS | **100.0** [88.6, 100.0] |
| | BTS | 93.3 [78.7, 98.2] |
| | MENTS | 90.0 [74.4, 96.5] |
| | MCTS-IS | 83.3 [66.4, 92.7] |
| | AlphaZero | 70.0 [52.1, 83.3] |
| Atari | MCTS-IS | **83.3** [66.4, 92.7] |
| | DENTS | **83.3** [66.4, 92.7] |
| | MaxMCTS | **83.3** [66.4, 92.7] |
| | MCTS | 80.0 [62.7, 90.5] |
| | MENTS | 80.0 [62.7, 90.5] |
| | DR | 70.0 [52.1, 83.3] |
| | BTS | 66.7 [48.8, 80.8] |

### F.2 TARGET POLICY ABLATION

Table 8 compares alternative target policy formulations for the DR estimator. Softmax consistently outperforms $\epsilon$-greedy, visit-based, and UCB-based alternatives.

Table 8: Target policy ablation on Go (100 rollouts). (a) Win rates of MCTS-DR-$\beta^*$ with softmax target vs. alternatives. (b) Elo ratings.

<table>
<tr><td colspan="2" align="center">(a) Win Rates</td><td colspan="2" align="center">(b) Elo Ratings</td></tr>
<tr><td>Target Policy</td><td>Win Rate (%)</td><td>Target Policy</td><td>Elo</td></tr>
<tr><td>vs. $\epsilon$-greedy</td><td>100.0 [88.6, 100.0]</td><td>Softmax ($\beta^*$)</td><td>1906.2</td></tr>
<tr><td>vs. Visit-based</td><td>100.0 [88.6, 100.0]</td><td>$\epsilon$-greedy</td><td>1686.9</td></tr>
<tr><td>vs. UCB-based</td><td>100.0 [88.6, 100.0]</td><td>Visit-based</td><td>1394.6</td></tr>
<tr><td></td><td></td><td>UCB-based</td><td>1012.3</td></tr>
</table>

### F.3 VARIANCE-BASED VS. FALLBACK $\beta$ USAGE

Table 9 shows how often MCTS-DR successfully computes the variance-minimizing $\beta^*$ versus falling back to the heuristic $\beta = 0.5$ due to insufficient samples. The higher fallback rate in Go reflects its larger branching factor, which spreads visits more thinly across actions.

Table 9: Distribution of $\beta$ computation methods across domains

| Domain | Variance-based $\beta^*$ (%) | Fallback (%) |
|---|---|---|
| Go | 56.5 | 43.5 |
| VirtualHome | 88.8 | 11.2 |

## G CODE AVAILABILITY AND LICENSE

We commit to releasing our full implementation upon acceptance, which will include:

- Complete implementations of DR-MCTS, IS-MCTS, and baseline MCTS algorithms
- Environment wrappers for Go, Atari, GSM8K, and VirtualHome
- Reproduction scripts for all experiments
- Hyperparameter configurations and random seeds for reproducibility
- Detailed documentation and usage instructions

Our implementation builds upon the following open-source resources:

- **VirtualHome Environment**: Based on Watch-and-Help[1] (CC BY-NC-SA 4.0 license) for training/test data generation
- **LLM-MCTS Integration**: Adapted from Zhao et al.'s codebase[2] (CC BY-NC-SA 4.0 license) for LLM-guided tree search

All code will be released under the MIT license to facilitate broader adoption. We acknowledge that our experiments utilized GPT-4o and GPT-4o-mini APIs; users will need their own API credentials to reproduce GSM8K and VirtualHome results.

### G.1 REPRODUCIBILITY GUIDELINES

To reproduce our experiments:

---

[1]https://github.com/xavierpuigf/virtualhome/tree/master
[2]https://github.com/1989Ryan/llm-mcts

- $5 \times 5$ **Go**: Any modern CPU with at least 1GB RAM
- **GSM8K**: GPU GPU with minimum 16GB memory (A100 recommended for exact timing replication)
- **VirtualHome**: GPU with minimum 16GB memory (A100 recommended for exact timing replication)
- **Software requirements**: Python 3.8+, PyTorch 1.10+, and API access to GPT-4o and GPT-4o-mini
- **Estimated total compute time**:
  - $5 \times 5$ Go: up to 30 hours for all experiments
  - Atari: up to 30 hours for all experiments
  - GSM8K: Up to 100 hours
  - VirtualHome: Up to 100 hours

