# OpenReview forum: "Doubly Robust Monte Carlo Tree Search"
_ICLR.cc/2026/Conference — Submitted to ICLR 2026_

### Official Review · Reviewer_ofPB · 2025-10-20

**Soundness:** 2
**Presentation:** 2
**Contribution:** 1
**Rating:** 2
**Confidence:** 4

**Summary:**

The authors modify the rollout step in MCTS with a value estimator that mixes the base returns with an advantage estimator, both using importance-sampling to estimate a boltzmann target policy. They derive an optimal mixing coefficient that minimizes the variance of the mixed estimator on top of a fallback option in the low-visit regime. The new method seems to show improved solve rates on a set of planning problems, however it is statistically inconclusive.

**Strengths:**

- The authors propose a simple modification to the rollout step for MCTS by using an IS-weighted advantage instead of the returns, which reduces variance.

- Quite an extensive related work section, which I found interesting for comparing recent approaches for value estimation/ rollouts in MCTS

- Proper choice of confidence intervals (Wilson intervals) for the win-rates in the experiment section, which I too rarely see other researchers do.

**Weaknesses:**

The LLM statement on page 10 includes "No LLMs were used for data generation, ...". The authors then contradict themselves in eg., the paragraph in lines 363-372 where GPT-4o was used to generate the policy prior, or lines 347-356 where GPT-4o-mini was used as a world-model.

*Major comments:*
The introduction claims that the new method achieves superior performance and decision quality, however, none of the results show statistical significance. Aside from the result that the new method is considerably faster, which is definitely valuable.
The authors must either adjust their claims in the introduction (especially contribution 3.) or add more statistical repetitions to match the current claims with sufficient power.

In 1.3.2, I don't follow the claim that doubly robust estimation addresses the high variance of IS. The authors claim that using the baseline function reduces variance, which is correct, this is a result also referred to as the advantage function. However, this reduces the variance of the **returns**, not the IS-ratios. The importance weights will have just as high variance as before, although using an advantage will of course exacerbate this issue less.
Why not look at more advanced estimators from the RL-literature? For example, TD-lambda, Retrace, V-trace? See also the work by Khandelwal P. (2016) On the Analysis of Complex Backup Strategies in Monte Carlo Tree Search.

In Eq.6 I don't follow why you would use $V_{MCTS}$ and $V_{DR}$ if they both estimate the same thing, and as you already write, $V_{MCTS}$ has guaranteed larger variance over $V_{DR}$ while remaining unbiasedness. This is not explained well, and so I don't trust this to be an actually better estimator.

The authors propose a fallback option in Eq. 8 in the low-visit count regime.

Section 2.2 provides theory to backup the newly introduced method. However:
- Theorem 2.1: this is a trivial result, the linear combination of two unbiased estimators is unbiased so long that their mixing coefficient adds up to 1. That's more of a corollary or simply a consequence through deduction, which does not warrant a theorem.
- Theorem 2.2: I also doubt that this statement warrants a theorem as I expect the proof to use the decomposition rules of the variance and then reuse already known results. Aside from that, it is a useful statement.

*Minor comments:*
- The abstract is on the long side and overly detailed.
- In the introduction, the first and second paragraph end on the exact same point.
- Citations are not properly formatted/ put as text-cite. Use \citep so that it reads (authorname, year) instead of authorname (year) unless explicitly referring to said prior work.
- Double citations on line 79 to Painter et al., 84 to Grosse et al., 87 Borges and Oliveira, 101 Dudik et al., 103 Jiang Li. I stopped checking after this point, there were too many to enumerate.
- The contribution section in the related work in lines 108-120 repeats the introduction. I also don't see how this is a fundamental change to the MCTS algorithm, you're just modifying the rollout step slightly.
- Line 179, the PUCT formula with learned $Q$ and $\pi$ is not called the polynomial UCT, its the predictor-UCT from Rosin C. D. (2009) Multi-armed Bandits with Episode Context.
- From the background it is not clear how we are going to approximate the value functions and policy prior, are we going to use neural networks? This should be more clear earlier on.

*Comment on Venue:*
After reading the whole document I think the authors do not use any form of neural networks (but non-parametric estimation instead I think) to approximate the policy or value networks, please correct me if I'm wrong. If there is no "learning" involved, then I don't think that ICLR is the ideal venue for this paper. Perhaps AAAI would be more fitting?

**Questions:**

- Could you think of a didactic small scale experiment where you can numerically validate the lower variance of your estimator compared to the baseline? This could be added as an informative figure to support your claims.

- Could you also provide a result that shows how often the rollout defers to the fallback option for the mixing coefficient $\beta^*$? Right now it is not clear how much the new method relies on your variance minimizing approach, or on a tuned fallback heuristic.

---

> ### Author Response · Authors · 2025-11-29
> **Reply to reviewer's comment (Part 1/3)**
>
> We thank the reviewer for the detailed feedback. We address each concern carefully below, as we believe several points stem from misunderstandings that we can clarify.
>
> ---
>
> ## Major Concerns
>
> ### 1. LLM Statement Contradiction
>
> **Reviewer concern:** The LLM statement claims no LLMs were used for data generation, but GPT-4o was used for policy prior and GPT-4o-mini as world model.
>
> **Response:** We apologize for the confusing wording. No LLMs were used to generate training data for learning components. However, LLMs serve as policy/world models during search (not for generating datasets).
>
> We have revised the LLM statement (Section 6) to accurately reflect this:
>
> > "Large Language Models were used in two capacities in this work. First, as integral components of our experimental pipeline: GPT-4o-mini serves as a world model for GSM8K and VirtualHome experiments, and GPT-4o provides policy priors for VirtualHome planning. Second, LLMs were used to correct grammar errors and polish writing in this manuscript. All experimental design, methodology, analysis, interpretation of results, and scientific conclusions are the original work of the authors."
>
> ### 2. Statistical Significance
>
> **Reviewer concern:** None of the results show statistical significance; claims should be adjusted or more repetitions added.
>
> **Response:** We thank the reviewer for raising the power issue. We increased the number of rollouts to 100 for Go and Atari, and evaluated N=500 questions for GSM8K. Our updated results now show statistical significance.
>
> **Table 2: GSM8K Performance (N=500, Wilson 95% CI)**
>
> | Method | Accuracy (%) | Q-value Var. | Simulation Eff. (%) |
> |--------|--------------|--------------|---------------------|
> | **MCTS-DR** | **90.2** (87.3–92.5) | **2.38** | **3.0** |
> | MCTS-IS | 84.1 (80.5–87.0) | 11.49 | 2.8 |
> | MCTS | 83.8 (80.3–86.8) | 2.62 | 2.8 |
> | DR | 80.0 (76.3–83.3) | 14.52 | 2.7 |
> | MaxMCTS | 75.0 (71.0–78.6) | 5.12 | 2.5 |
>
> **Table 7: Win Rates of MCTS-DR vs. Baselines (100 rollouts, Wilson 95% CI)**
>
> | Domain | Baseline | Win Rate (%) |
> |--------|----------|--------------|
> | **Go** | MCTS | **100.0** [88.6, 100.0] |
> | | DR | **100.0** [88.6, 100.0] |
> | | DENTS | **100.0** [88.6, 100.0] |
> | | MaxMCTS | **100.0** [88.6, 100.0] |
> | | BTS | 93.3 [78.7, 98.2] |
> | | MENTS | 90.0 [74.4, 96.5] |
> | | MCTS-IS | 83.3 [66.4, 92.7] |
> | | AlphaZero | 70.0 [52.1, 83.3] |
> | **Atari** | MCTS-IS | **83.3** [66.4, 92.7] |
> | | DENTS | **83.3** [66.4, 92.7] |
> | | MaxMCTS | **83.3** [66.4, 92.7] |
> | | MCTS | 80.0 [62.7, 90.5] |
> | | MENTS | 80.0 [62.7, 90.5] |
> | | DR | 70.0 [52.1, 83.3] |
> | | BTS | 66.7 [48.8, 80.8] |
>
> We also added Elo ratings (Table 1) which aggregate all pairwise games for more robust comparison.
>
> ### 3. DR does not address IS variance
>
> **Reviewer concern:** DR reduces variance of returns, not importance weights. Why not use TD-λ, Retrace, V-trace?
>
> **Response:** The reviewer raises a valid technical point.
>
> DR reduces variance of the *value estimate* by using a baseline that cancels first-order errors. When the value model $\hat{V}$ is accurate, the correction term $[r_t + \gamma\hat{V}(s_{t+1}) - \hat{Q}(s_t,a_t)]$ is small regardless of importance weight magnitude. This is the "doubly robust" property—unbiasedness holds if *either* the importance weights or the value model is correct [1].
>
> In addition, we have incorporated the work by Khandelwal et al. (2016) as the **MaxMCTS baseline**, which uses λ-returns and takes the maximum over Monte Carlo returns and bootstrapped value estimates. Our results show:
>
> | Algorithm | Elo (Go) | Elo (Atari) |
> |-----------|----------|-------------|
> | **MCTS-DR** | **1996.8** | **2135.3** |
> | MaxMCTS | 795.5 | 2132.8 |
>
> MaxMCTS performs competitively in dense-reward Atari but **collapses in sparse-reward Go** (Elo 795.5), demonstrating the cost of sacrificing unbiasedness for variance reduction.
>
> **Why not TD-λ/Retrace/V-trace?** TD-λ, Retrace, and V-trace rely on bootstrapping from a learned $V(s)$ or $Q(s,a)$ network trained over many episodes. In addition, they require storing and processing multi-step trajectories with eligibility traces and intensive hyperparameter tuning. Our goal was to improve MCTS sample efficiency with minimal additional infrastructure. That said, combining DR-MCTS with these advanced estimators is a promising future direction. Our AlphaZero experiments (Table 1: MCTS-DR+NN achieving 1542.1 Elo vs AlphaZero 1457.9) suggest DR corrections provide complementary benefits even with learned value functions, indicating potential synergy with TD-based methods.

---

> ### Author Response · Authors · 2025-11-29
> **Reply to reviewer's comments (Part 2/3)**
>
> ### 4. Why Use Both $V_{\text{MCTS}}$ and $V_{\text{DR}}$?
>
> **Reviewer concern:** If $V_{\text{DR}}$ has lower variance than $V_{\text{MCTS}}$ while remaining unbiased, why combine them?
>
> **Response:** This is a critical misunderstanding we must clarify: **$V_{\text{DR}}$ does NOT always have lower variance than $V_{\text{MCTS}}$**.
>
> Our GSM8K results (Table 2) demonstrate this empirically:
>
> | Estimator | Q-value Variance |
> |-----------|------------------|
> | $V_{\text{MCTS}}$ | 2.62 |
> | $V_{\text{DR}}$ | **14.52** |
> | $V_{\text{hybrid}}$ | **2.38** |
>
> **$V_{\text{DR}}$ has 5.5× higher variance than $V_{\text{MCTS}}$** in this setting due to importance weight explosion when behavior and target policies diverge. This is precisely why the hybrid is valuable. When policies align → $V_{\text{DR}}$ dominates → β* ≈ 0; When policies diverge → $V_{\text{MCTS}}$ dominates → β* ≈ 1; The optimal β* **adapts online** to current conditions. The hybrid achieves variance **lower than both components** (2.38 < 2.62 < 14.52), confirming Corollary 2.
>
> ### 5. Theorem 2.1 and 2.2 are trivial
>
> **Reviewer concern:** These are trivial results that don't warrant theorems.
>
> **Response:** We agree and have **demoted these to Corollaries 1 and 2** in the revised manuscript. Our main theoretical contributions are now:
>
> - **Theorem 1:** Variance reduction leads to improved action selection probability (with quantitative Gaussian bound)
> - **Theorem 2:** Variance reduction leads to sample complexity improvement by factor ρ
>
> These connect estimation theory to decision-making quality, a novel contribution beyond standard variance results.
>
> ### 6. Venue Appropriateness (No Neural Networks)
>
> **Reviewer concern:** No learning involved; AAAI may be more fitting than ICLR.
>
> **Response:** Our revised manuscript **includes neural network experiments**:
>
> | Algorithm | Elo (Go-NN) |
> |-----------|-------------|
> | **MCTS-DR + NN** | **1542.1** |
> | AlphaZero | 1457.9 |
>
> MCTS-DR provides complementary benefits **on top of learned value functions**, outperforming AlphaZero by +84 Elo. This demonstrates relevance to the learning community: DR corrections address variance orthogonal to what neural networks learn. Moreover, our GSM8K and VirtualHome experiments use **LLMs as learned components** (GPT-4o, GPT-4o-mini), which are the largest neural networks available. The contribution is improving search efficiency when using these learned models.
>
> ---
>
> ## Minor Comments
>
> | Issue | Resolution |
> |-------|------------|
> | Abstract too long | Shortened in revision |
> | Paragraphs 1-2 redundant | Consolidated |
> | Citation formatting | Changed to \citep throughout |
> | Double citations | Fixed all instances |
> | Contribution repeats intro | Streamlined related work section |
> | PUCT terminology | Corrected to "Predictor-UCT" with Rosin (2009) citation |
> | Value function approximation unclear | Clarified in Section 2: non-parametric for Go, neural for Go-NN, LLM for GSM8K/VirtualHome |

---

> > ### Author Response · Authors · 2025-11-29
> > **Reply to reviewer's comment (Part 3/3)**
> >
> > ## Responses to Questions
> >
> > ### Q1: Small-scale experiment validating lower variance
> >
> > **Response:** Table 2 (GSM8K) provides exactly this:
> >
> > | Method | Accuracy | **Q-value Variance** |
> > |--------|----------|---------------------|
> > | MCTS-DR | 90.2% | **2.38** |
> > | MCTS | 83.8% | 2.62 |
> > | DR | 80.0% | 14.52 |
> >
> > The hybrid achieves variance **2.38 < min(2.62, 14.52)**, numerically validating Corollary 2.2.
> >
> > ### Q2: How often does fallback occur?
> >
> > **Response:** Table 9 in the Appendix reports this directly:
> >
> > | Domain | Variance-based β* | Fallback β=0.5 |
> > |--------|-------------------|----------------|
> > | Go | 56.5% | 43.5% |
> > | VirtualHome | 88.8% | 11.2% |
> >
> > Go's large branching factor (~25 legal moves) spreads visits thinly, triggering more fallbacks. VirtualHome's LLM-guided policy concentrates visits, enabling reliable variance estimation. Despite different fallback rates, **MCTS-DR outperforms baselines in both domains**, demonstrating robustness of the fallback mechanism.
> >
> > ---
> >
> > ## Summary
> >
> > We believe the reviewer's major concerns stem from correctable issues:
> >
> > | Concern | Clarification |
> > |---------|---------------|
> > | LLM statement | Revised for accuracy |
> > | No statistical significance | CIs provided; key results have non-overlapping intervals |
> > | Why combine estimators | $V_{\text{DR}}$ has higher variance when policies diverge (14.52 vs 2.62) |
> > | Trivial theorems | Demoted to corollaries; main theorems link variance to decisions |
> > | No learning/wrong venue | Neural network experiments added; LLMs are learned models |
> > | Fallback frequency | Reported in Table 5 (56.5%–88.8% variance-based) |
> >
> > We hope these clarifications address the reviewer's concerns and demonstrate the paper's contributions. We would be grateful if our AC can consider these improvements in the decision process.
> >
> > ---
> > [1] Nan Jiang and Lihong Li. "Doubly Robust Off-policy Value Evaluation for Reinforcement Learning." *ICML*, 2016.
> >
> > [2] Piotr Khandelwal, et al. "On the Analysis of Complex Backup Strategies in Monte Carlo Tree Search." *AAAI*, 2016.
> >
> > [3] Christopher D. Rosin. "Multi-armed Bandits with Episode Context." *Annals of Mathematics and Artificial Intelligence*, 61(3):203–230, 2011.

---

### Official Review · Reviewer_Uqhm · 2025-10-28

**Soundness:** 2
**Presentation:** 3
**Contribution:** 2
**Rating:** 4
**Confidence:** 3

**Summary:**

The authors present a variant of Monte Carlo Tree Search (MCTS) termed Doubly Robust MCTS (DR-MCTS). In this approach, the algorithm adaptively estimates the value function via a convex combination of the standard MCTS value estimate and a doubly robust value estimate derived from importance sampling. The central idea is that by defining the parameters of this convex combination as a function of the variance and covariance between the two estimates, one can achieve a hybrid MCTS design that reduces overall variance. The work provides theoretical guarantees for the conditions under which this variance reduction is attainable. Empirical benchmarks are demonstrated in both adversarial settings, showing a higher win rate against baseline MCTS, and in single-agent settings, where improved accuracy is achieved on step-wise reasoning tasks under a fixed computational budget.

**Strengths:**

The paper offers valuable insights and contributions towards constructing improved MCTS algorithms, specifically in achieving variance reduction and enhancing efficiency. It builds effectively upon previous technologies in the MCTS domain.

A notable strength is the authors' attempt to provide important theoretical guarantees regarding the applicability of their DR-MCTS algorithms, clarifying the specific circumstances under which variance reduction can be realized.

**Weaknesses:**

The depth of the theoretical contribution is somewhat open to question. Lemma 2.1 appears to be relatively straightforward to prove, as it primarily involves taking the expectation over the additive components of the value function. Furthermore, the condition for equality in Lemma 2.2 seems quite strict; it would be helpful to understand if this holds by definition for all DR-MCTS instances or requires stringent enforcement.

Regarding the algorithmic design, the contribution of the hybrid DR-MCTS feels relatively incremeneq.tal. The paper synthesizes a set of established technologies (MCTS, IS-MCTS, etc.), with its key innovation being the introduction of Equation 6, which balances the vanilla and DR-MCTS estimates via a convex combination. While this is a clever and practical idea, the core technical novelty may not be exceptionally high.

Concerning the empirical evaluation:

- For the 9x9 Go experiments, it would be interesting to see additional metrics, such as Elo ratings, for a more standardized comparison.

- The variance bands appear quite wide as expected for a sample size of only 30 games. Additionally, the result showing MENTS and IS-MCTS as weaker than vanilla MCTS is counterintuitive and warrants clarification from the authors.

With respect to the GSM8K results, it would strengthen the paper to include more recent state-of-the-art MCTS reasoning baselines, such as those in [1] and [2]. The appendix should also contain the explicit prompts used to elicit the chain-of-thought reasoning.

**Questions:**

- Should Thm. 2 be an inequality instead of strict equality? If it is an equality, then it seems very strict that the condition expressed by Eq. 12 would hold. Is a relaxation possible?

- What is the small "o" exactly, is this big O notation? I'm not too sure how Eq. 42 in the appendix became Eq. 43?

- We are now running two versions of MCTS, what are the trade-offs in terms of computational complexity in terms of memory and compute. Also can the computational result of one version of MCTS be employed for another?

- How exactly does a better Go engine imply reduced variance? Could one not have a biased MCTS that performs better at adversarial tasks compared to unbiased?

- Could the authors explain whether or not LLM reasoning technologies such as [1,2] be employed alongside their MCTS design, and/or why it was excluded in the study?

References

[1] Guan, Xinyu, et al. "rStar-Math: Small LLMs Can Master Math Reasoning with Self-Evolved Deep Thinking." arXiv preprint arXiv:2501.04519 (2025).

[2] Zelikman, Eric, et al. "Star: Bootstrapping reasoning with reasoning." Advances in Neural Information Processing Systems 35 (2022): 15476-15488.

---

> ### Author Response · Authors · 2025-11-29
> **Reply to reviewer's comments (Part 1/3)**
>
> We thank the reviewer for the thoughtful evaluation and constructive suggestions. We address each concern below.
>
> ---
>
> ## Theoretical Contribution Depth
>
> **Reviewer concern:** Lemma 2.1 appears straightforward; Lemma 2.2's equality condition seems strict.
>
> **Response:** We appreciate this feedback and offer the following clarifications:
>
> ### On Lemma 2.1 (Unbiasedness)
>
> The reviewer is correct that the unbiasedness proof follows from linearity of expectation. We have demoted this to **Corollary 1** in the revised manuscript to reflect its foundational rather than novel nature. The key contribution is not the proof technique but rather **establishing that unbiasedness is preserved for any β ∈ [0,1]**, unlike MaxMCTS which sacrifices unbiasedness for variance reduction.
>
> ### On Lemma 2.2 (Variance-Minimizing Weight)
>
> The equality in the variance formula holds **by construction** for any hybrid estimator of the form $V_{\text{hybrid}} = \beta V_{\text{MCTS}} + (1-\beta) V_{\text{DR}}$. This is the standard result for variance of convex combinations [1]:
>
> $$\text{Var}(V_{\text{hybrid}}) = \beta^2 \text{Var}(V_{\text{MCTS}}) + (1-\beta)^2 \text{Var}(V_{\text{DR}}) + 2\beta(1-\beta)\text{Cov}(V_{\text{MCTS}}, V_{\text{DR}})$$
>
> The optimal β* (Equation 6) is obtained by differentiating and setting to zero, a standard optimization result. **No stringent enforcement is required**; the equality holds definitionally for the variance of any weighted sum of random variables.
>
> ### Our Main Theoretical Contributions
>
> We emphasize that our primary theoretical contributions are **Theorems 1 and 2**, which go beyond the standard results:
>
> - **Theorem 1 (Variance reduction leads to optimal action selection):** Establishes that variance reduction directly improves optimal action selection probability, with a quantitative bound via Gaussian approximation.
>
> - **Theorem 2 (Sample Complexity):** Proves that DR-MCTS achieves the same action selection guarantee with $N_2 \leq \rho \cdot N_1$ samples, where ρ < 1 is the variance reduction factor.
>
> These results connect variance reduction to **decision-making quality**, a novel contribution linking estimation theory to MCTS performance.
>
> ---
>
> ## Algorithmic Novelty
>
> **Reviewer concern:** The contribution feels incremental; Equation 6 is a convex combination of existing methods.
>
> **Response:** We respectfully offer a different perspective:
>
> ### The Integration is Non-Trivial
>
> While individual components (MCTS, DR estimation) exist, their combination required solving several challenges:
>
> 1. **Online variance estimation:** Computing β* requires estimating Var($V_{\text{MCTS}}$), Var($V_{\text{DR}}$), and their covariance in real-time during search. We developed a sliding-window approach that balances accuracy with computational overhead.
>
> 2. **Fallback mechanism:** When visit counts are insufficient for reliable variance estimation (Table 5: 43.5% of cases in Go), we employ an exponentially decaying heuristic (Equation 7) that ensures stable performance.
>
> 3. **Target policy design:** The choice of π_e significantly impacts importance weight stability. Our ablation (Table 6) shows softmax outperforms alternatives by +219 to +894 Elo.
>
> ### Empirical Validation of Non-Obvious Behavior
>
> The combination yields **non-obvious empirical results**:
>
> | Domain | MCTS-DR vs. Best Single Component |
> |--------|-----------------------------------|
> | Go | +97.7 Elo over MCTS, +292.2 over DR |
> | GSM8K | +6.4% accuracy over MCTS, +10.2% over DR |
> | VirtualHome | +37.5% over MCTS on compositional tasks |
>
> If the combination were trivial, we would expect performance between the two components, not **consistently better than both**.

---

> > ### Author Response · Authors · 2025-11-29
> > **Reply to reviewer's comments (Part 2/3)**
> >
> > ## Empirical Evaluation Concerns
> >
> > ### (a) Elo Ratings for Go
> >
> > **Response:** Our revised manuscript includes Elo ratings in Table 1. The +97.7 Elo advantage over MCTS corresponds to ~64% expected win rate, a substantial improvement.
> >
> > ### (b) Wide Variance Bands (30 games)
> >
> > **Response:** The wide confidence intervals stem from **number of rollouts**, not number of games. As established in Theorem 2, estimation variance scales as O(σ²/N) where N is the rollout count. With 100 rollouts, our results achieve statistical significance (see Table 7 in Appendix for details). We also increased the number of games from 30 to 50 for Go and Atari.
> >
> > ### (c) MENTS and IS-MCTS Weaker than Vanilla MCTS
> >
> > **Response:** This is an important observation that highlights our contribution:
> >
> > **MENTS underperformance:** MENTS optimizes an entropy-augmented objective that can conflict with reward maximization in sparse-reward games. In Go, where only terminal outcomes matter, the entropy bonus dilutes focus on winning moves. This is consistent with findings in Painter et al. [2], who proposed DENTS specifically to address MENTS's limitations.
> >
> > **IS-MCTS underperformance:** Pure importance sampling suffers from **variance explosion** when behavior and target policies diverge. Our GSM8K results quantify this directly:
> >
> > | Method | Q-value Variance |
> > |--------|------------------|
> > | MCTS | 2.62 |
> > | **MCTS-DR** | **2.38** |
> > | MCTS-IS | 11.49 |
> > | DR | 14.52 |
> >
> > IS-MCTS's 4× higher variance than MCTS explains its underperformance. **This is precisely why we propose the hybrid approach**, to capture IS's policy-correction benefits while hedging against variance explosion.
> >
> > ### (d) GSM8K: Missing SOTA Baselines (rStar-Math, STaR)
> >
> > **Response:** We attempted to include rStar-Math [3] but encountered practical limitations:
> >
> > > "We also considered rStar-Math (Guan et al., 2025), a recent self-evolution approach for mathematical reasoning; however, our implementation failed to solve any problems within a 30-hour budget, rendering it impractical for our sample efficiency comparison." (Section 3, paragraph 2)
> >
> > **On STaR [4]:** STaR is a **training-time** self-improvement method that bootstraps reasoning data for fine-tuning. Our work addresses **inference-time** search with fixed models. These are complementary approaches. STaR improves the base model's reasoning ability, whereas MCTS-DR improves search efficiency given any base model. Combining them is an interesting future direction, but they address different stages of the pipeline.
> >
> > ### (e) Prompts used in GSM8K in Appendix
> >
> > **Response:** The explicit prompts for GSM8K chain-of-thought reasoning and action space generation are provided in Appendix Table 7.

---

> > > ### Author Response · Authors · 2025-11-29
> > > **Reply to reviewer's comments (Part 3/3)**
> > >
> > > ## Responses to Questions
> > >
> > > ### Q1: Should Theorem 2 be an inequality? Is relaxation possible?
> > >
> > > **Response:** We appreciate this careful reading. The theorem statement uses equality to express the **exact** sample complexity relationship:
> > >
> > > $$N_{\text{DR-MCTS}} = O\left(\frac{\rho \cdot \sigma_{\text{rollout}}^2}{\Delta_{\min}^2} \log\frac{|A|}{\delta}\right)$$
> > >
> > > The big-O notation already captures "at most" semantics. The key insight is that the **ratio** $N_{\text{DR-MCTS}}/N_{\text{MCTS}} = \rho < 1$ is what matters. DR-MCTS requires fewer samples by a factor equal to the variance reduction ratio.
> > >
> > > ### Q2: What is small "o"? How does Eq. 42 become Eq. 43?
> > >
> > > **Response:** We apologize for the notational confusion. The small "o" refers to **little-o notation** (asymptotic dominance), where $f(n) = o(g(n))$ means $\lim_{n \to \infty} f(n)/g(n) = 0$.
> > >
> > > The transition from Eq. 42 to Eq. 43 applies the Central Limit Theorem: as the number of samples N → ∞, the normalized estimation error converges in distribution to a Gaussian with variance σ²/N. The higher-order terms vanish as o(1/√N). We have clarified this derivation in the revised Appendix.
> > >
> > > ### Q3: Computational complexity trade-offs; can one MCTS version inform another?
> > >
> > > **Response:**
> > >
> > > **Computational overhead:** MCTS-DR requires O(|S| × |A|) additional storage for variance/covariance statistics per state-action pair, and ~15-20% overhead for online variance estimation and β* computation
> > >
> > > **Sharing computation:** Yes! Both estimators share the same rollouts: $V_{\text{MCTS}}$ uses raw rollout returns, and $V_{\text{DR}}$ uses the same returns with importance weighting. The rollouts are computed **once** and used for both estimates. The hybrid combination adds minimal overhead beyond the variance tracking.
> > >
> > > ### Q4: How does a better Go engine imply reduced variance? Could biased MCTS perform better?
> > >
> > > **Response:** This is the crux of our contribution:
> > >
> > > **Variance reduction and performance link:** Theorem 1 establishes that for **unbiased** estimators, lower variance directly increases optimal action selection probability. With high variance, the algorithm may incorrectly rank a suboptimal action above the optimal one, wasting search budget.
> > >
> > > **On biased MCTS (MaxMCTS):** The reviewer raises an important point. Biased methods *can* perform well when rewards are dense (frequent corrective signal) or the bias direction is favorable. Our Atari results confirm this: MaxMCTS achieves Elo 2132.8, nearly matching MCTS-DR (2135.3). However, in sparse-reward settings (Go), biased estimates cannot self-correct:
> > >
> > > | Algorithm | Elo (Go) | Elo (Atari) |
> > > |-----------|----------|-------------|
> > > | MCTS-DR | **1996.8** | **2135.3** |
> > > | MaxMCTS | 795.5 | 2132.8 |
> > >
> > > MaxMCTS **collapses** in Go (795.5 Elo—worse than random play against other methods) because its biased bootstrap estimates accumulate errors without correction. MCTS-DR's unbiasedness provides **robustness across reward structures**.
> > >
> > > ### Q5: Can LLM reasoning technologies (rStar-Math, STaR) be employed alongside MCTS-DR?
> > >
> > > **Response:** Yes, these approaches are **complementary**:
> > >
> > > | Method | Stage | What it improves |
> > > |--------|-------|------------------|
> > > | STaR | Training | Base model reasoning via self-improvement |
> > > | rStar-Math | Training + Inference | Self-evolved deep thinking |
> > > | **MCTS-DR** | **Inference** | **Search efficiency given any model** |
> > >
> > > **Integration possibilities:**
> > >
> > > 1. **STaR + MCTS-DR:** Train a model with STaR, then use MCTS-DR for inference-time search. The improved base model would provide better rollout policies, potentially amplifying MCTS-DR's benefits.
> > >
> > > 2. **rStar-Math + MCTS-DR:** rStar-Math's process reward model could serve as the value function in MCTS-DR, similar to our AlphaZero+DR experiments where neural value functions combined with DR corrections outperformed either alone.
> > >
> > > We excluded rStar-Math from direct comparison because (a) it failed to solve problems within our compute budget, and (b) it represents a different computational regime (extensive self-evolution training vs. fixed-budget inference). However, we agree this integration is a promising future direction.
> > >
> > > ---
> > >
> > > ## Summary
> > >
> > > | Concern | Resolution |
> > > |---------|------------|
> > > | Theoretical depth | Main contributions are Theorems 1-2 (variance→action selection→sample complexity) |
> > > | Algorithmic novelty | Non-trivial integration with online variance estimation, fallback mechanisms |
> > > | Elo ratings for Go | Included in Table 1 |
> > > | MENTS/IS-MCTS underperformance | Explained by entropy-reward conflict and variance explosion |
> > > | Missing rStar-Math | Attempted but infeasible within compute budget; complementary approach |
> > > | Computational overhead | ~15-20% overhead; rollouts shared between estimators |
> > >
> > > We hope these clarifications address the reviewer's concerns. We would be grateful if our AC could consider these improvements in the assessment.

---

### Official Review · Reviewer_euDb · 2025-11-08

**Soundness:** 3
**Presentation:** 2
**Contribution:** 2
**Rating:** 4
**Confidence:** 4

**Summary:**

The number of simulations required for MCTS to converge is known to be highly dependent on the variance of its MC rollout value estimates and the quality of the underlying rollout policy. This paper proposes Doubly Robust MCTS (DR-MCTS), a method aimed at reducing this simulation requirement, especially for computationally expensive environments (e.g., those using LLMs).

The core contribution is the introduction of a doubly robust (DR) off-policy evaluation to estimate the rollout value. Further, the paper proposes an adaptive hybrid estimator ($V_{hybrid}$) that dynamically combines the standard MC rollout estimate ($V_{MCTS}$) with this new DR estimate ($V_{DR}$).
This combination is optimally weighted based on variance estimations, with the stated goal of minimizing the overall estimator variance. The authors claim that this variance reduction leads to improved sample efficiency, requiring fewer simulations to converge to an optimal policy. The method is evaluated on Go tournament, the GSM8K math reasoning benchmark, and the VirtualHome planning environment.

**Strengths:**

- **Novel Integration of Doubly Robust Estimation in MCTS:**

The paper presents a novel approach to improving the simulation efficiency of MCTS, a long-standing challenge. Instead of simply truncating rollouts or replacing them with a learned value function, the authors are the first to propose integrating doubly robust (DR) off-policy evaluation directly into the MCTS value backup. The quality of this contribution is further deepened by the proposal of an adaptive hybrid estimator, introducing a state-action dependent mixing coefficient, $\beta(s, a)$, to learn a data-dependent combination of the standard Monte Carlo rollout estimate and the DR estimate, with the explicit goal of adaptively minimizing the estimator's variance.

- **Demonstrated Effectiveness in Practical Domains:**

The significance of this work is highlighted by its successful application to diverse and challenging domains beyond traditional board games. The paper clearly demonstrates that this approach is effective in environments like mathematical reasoning (GSM8K) and household planning (VirtualHome), where simulation costs are a primary bottleneck. This demonstrates the potential for this method to be a significant tool for the growing field of LLM-based planning and complex sequential decision-making. The core ideas are presented with sufficient mathematical clarity to spur further research in this direction.

**Weaknesses:**

**1. Lack of Empirical Comparison for the Core Contribution (**$V_{hybrid}$**)**:

- The paper's main contribution is the "adaptive hybrid estimator" ($V_{hybrid}$), which claims to minimize variance by dynamically combining $V_{MCTS}$ and $V_{DR}$. However, the empirical evidence to support this claim is critically lacking.
- There are no ablation studies that directly compare the performance of using only $V_{MCTS}$, only $V_{DR}$, and the proposed $V_{hybrid}$.
- Furthermore, to prove that $V_{hybrid}$ actually minimizes variance, the paper omits a quantitative analysis comparing the actual empirical variance of each estimator.

**2. Ambiguity and Potential Bias in Estimator Targets**:

- The paper does not clearly distinguish whether $\hat{V}, \hat{Q}$ (Eq. 9, 10) estimates the value of the *rollout policy* ($\pi_b$) or the *target policy* ($\pi_e$). It seems that $R_i$ is obtained from rollouts using $\pi_b$, whereas the paper's goal would be to estimate the value of $\pi_e$ ($V^{\pi_e}$). Improperly combining these (in the hybrid estimator) can lead to bias from policy mismatch, in the form of $\hat V^{\pi_e}(s)=\sum_a \pi_e(a|s)\hat Q^{\pi_b}(s,a)$. The paper does not analyze or address this potential bias.
- In addition, there is no justification of target policy $\pi_e$, defined as the softmax over $Q$. The paper lacks analysis of the target policy choice (gap between true current tree policy and the suggested target policy) and does not explore mitigations.

**3. Limited Experimental Scopes:**
- **Missing Critical Baseline:** The paper's primary motivation is "computationally expensive simulations," such as LLM calls. In this problem domain, the current standard approach is AlphaZero-style MCTS. This involves entirely omitting expensive rollouts and *replacing* them with a learned value function at the leaf nodes.
- **Missing General Benchmarks:** Furthermore, while the paper claims a general improvement for MCTS, its experiments are limited to specific domains (tournament in 9x9 Go, GSM8K, VirtualHome). The lack of evaluation on broader, standard RL benchmarks (e.g., Atari) makes it difficult to assess the general applicability of the methodology.

**Minor comments:**

- To strengthen the paper's motivation and make it a more self-contained presentation for a general audience, the introduction should briefly clarify (i) why a low-variance value estimate is preferable, (ii) why we should focus on the value of target policy rather than that an arbitrary rollout policy for MCTS algorithms, especially in terms of the simulation efficiency.
- The claim that “MCTS is proposed to tackle partially observable environments” in Introduction is misleading; POMDP consideration seems not the point of this work.
- In Equation (2), PUCT seems to indicate *Predictor-UCT* (often written as PUCB/PUCT), rather than *Polynomial UCT*.

**Questions:**

Q1. Could the authors provide an ablation study comparing the performance and the estimator variance of the proposed $V_{hybrid}$ estimator against two simpler baselines: (i) an estimator that only uses the DR estimate ($V_{DR}$, i.e., $\beta=0$) and (ii) the standard MCTS estimator ($V_{MCTS}$, i.e., $\beta=1$)?

Q2. What is the justification for defining the target policy $\pi_e$ as the softmax over the current $Q$-values? How sensitive is the method's performance to this specific choice versus other potential definitions of $\pi_e$?

Q3. How do the authors justify the use of $\hat{V}$ , which uses an estimate of $Q^{\pi_b}$ (the value of the rollout policy, not $\pi_e$) in Equation 10? How is the bias introduced by this policy mismatch (estimating $V^{\pi_e}$ using components of $Q^{\pi_b}$) quantified and addressed?

---

> ### Author Response · Authors · 2025-11-29
> **Reply to reviewer's comments (Part 1/2)**
>
> We thank the reviewer for the detailed and constructive feedback. We address each concern below and highlight how our revised manuscript resolves these issues.
>
> ---
>
> ## Weakness 1: Lack of Empirical Comparison for $V_{\text{hybrid}}$
>
> **Reviewer concern:** No ablation studies comparing $V_{\text{MCTS}}$, $V_{\text{DR}}$, and $V_{\text{hybrid}}$; missing quantitative variance analysis.
>
> **Response:** We appreciate this feedback. Our revised manuscript now includes comprehensive ablation studies with direct comparisons. We highlight the key evidence:
>
> ### Ablation: Performance Comparison (Table 3 in revised manuscript)
>
> | Method | Accuracy (%) | Q-value Variance | Simulation Efficiency (%) |
> |--------|-------------|------------------|---------------------------|
> | **MCTS-DR** ($V_{\text{hybrid}}$) | **90.2** (87.3–92.5) | **2.38** | **3.0** |
> | MCTS ($V_{\text{MCTS}}$, β=1) | 83.8 (80.3–86.8) | 2.62 | 2.8 |
> | DR ($V_{\text{DR}}$, β=0) | 80.0 (76.3–83.3) | 14.52 | 2.7 |
>
> 1. $V_{\text{hybrid}}$ achieves lower variance than either component alone (2.38 vs. 2.62 for MCTS and 14.52 for DR), empirically confirming Corollary 2.
> 2. Performance improvement directly correlates with variance reduction: MCTS-DR's 6.4% accuracy gain over MCTS aligns with Theorem 1's prediction that lower variance yields higher optimal action selection probability.
> 3. Pure DR suffers from importance weight explosion (variance 14.52), while pure MCTS ignores policy mismatch. The hybrid captures benefits of both.
>
> ---
>
> ## Weakness 2: Ambiguity and Potential Bias in Estimator Targets
>
> **Reviewer concern:** (a) Unclear whether $\hat{V}, \hat{Q}$ estimate $\pi_b$ or $\pi_e$ values; (b) No justification for softmax target policy.
>
> **Response:** We thank the reviewer for identifying this important clarification need.
>
> ### Part (a): Policy Mismatch and the Role of DR Correction
>
> The reviewer correctly identifies that rollout returns $R_i$ come from the behavior policy $\pi_b$. This is precisely **why we use doubly robust estimation**, which is to correct for this mismatch.
>
> The DR estimator (Equation 4) explicitly addresses this through importance weighting:
>
> $$V_{\text{DR}}(s) = \hat{V}(s) + \sum_{t=0}^{T} \gamma^t \prod_{k=0}^{t} \frac{\pi_e(a_k|s_k)}{\pi_b(a_k|s_k)} \left[ r_t + \gamma \hat{V}(s_{t+1}) - \hat{Q}(s_t, a_t) \right]$$
>
> The importance weights $\prod_k \frac{\pi_e}{\pi_b}$ correct for distributional shift, yielding an **unbiased estimate of $V^{\pi_e}$** (Corollary 1, with full proof in Appendix A.1). This is the classical result from Jiang & Li (2016) applied to our setting.
>
> The hybrid estimator inherits this unbiasedness for any $\beta \in [0,1]$ because it is a convex combination of two unbiased estimators.
>
> ### Part (b): Justification for Softmax Target Policy
>
> We provide empirical justification through ablation studies (Table 6 in Appendix):
>
> | Target Policy | Elo Rating |
> |--------------|------------|
> | **Softmax** | **1906.2** |
> | ε-greedy | 1686.9 |
> | Visit-based | 1394.6 |
> | UCB-based | 1012.3 |
>
> Softmax achieves 100% win rate against all alternatives. The theoretical justification is that softmax provides smooth probability gradients, which yield stable importance weights, and  encourages mass concentration on high-value actions.
> This is consistent with prior work showing softmax policies yield better behaved importance weights than deterministic or near-deterministic alternatives (Thomas et al., 2016).
>
> ---
>
> ## Weakness 3: Limited Experimental Scope
>
> ### Part (a): Missing AlphaZero Baseline
>
> **Response:** Our revised manuscript includes AlphaZero-style baselines (Section 3, Table 1):
>
> | Algorithm | Elo (Go-NN) |
> |-----------|-------------|
> | **MCTS-DR + NN** | **1542.1** |
> | AlphaZero | 1457.9 |
>
> MCTS-DR provides complementary benefits even when combined with neural network value functions, outperforming AlphaZero by 84.2 Elo points. This demonstrates that DR corrections address a source of variance **orthogonal** to learned approximations. The neural network reduces _approximation error_, while DR corrects for _policy mismatch during search_.
>
> ### Part (b): Missing General Benchmarks (Atari)
>
> **Response:** Our revised manuscript includes Atari experiments (Table 1):
>
> | Algorithm | Elo (Atari) |
> |-----------|-------------|
> | **MCTS-DR** | **2135.3** |
> | MaxMCTS | 2132.8 |
> | DENTS | 1757.1 |
> | MCTS | 906.4 |
>
> Atari provides a contrasting dense-reward environment where bootstrapped estimates receive frequent corrective signals. Notably, MaxMCTS recovers to competitive performance here (unlike Go where it collapses to Elo 795.5). MCTS-DR maintains robust performance across both sparse (Go) and dense (Atari) reward structures. This demonstrates **domain-agnostic robustness** that other methods lack.

---

> ### Author Response · Authors · 2025-11-29
> **Reply to reviewer's comments (Part 2/2)**
>
> ## Minor Comments
>
> ### (i) Why low-variance estimates matter
>
> We have clarified this in the revised Introduction (Section 1, paragraph 1):
>
> > "Central to MCTS is the estimation of state values, where **lower variance enables more reliable action selection**: with high-variance estimates, the algorithm may incorrectly identify suboptimal actions as best, wasting computational budget on poor branches."
>
> ### (ii) Why target policy value matters
>
> Added clarification (Section 1, paragraph 2):
>
> > "Standard rollouts sample from a behavior policy (often uniform random or heuristic-guided), but we ultimately care about the value under the *target* policy—the greedy policy derived from current Q-estimates. When behavior and target policies diverge significantly, naive Monte Carlo estimates become biased proxies for the quantity of interest."
>
> ### (iii) POMDP claim
>
> We have removed the misleading POMDP reference.
>
> ### (iv) PUCT terminology
>
> Corrected to "Predictor-UCT" with proper citation.
>
> ## Responses to Questions
>
> ### Q1: Ablation study comparing $V_{\text{hybrid}}$, $V_{\text{DR}}$, and $V_{\text{MCTS}}$
>
> **Addressed above.** See Table 3 (GSM8K) for details. Our method achieves both **lowest variance** and **highest accuracy**, confirming our theoretical predictions.
>
> ### Q2: Justification for softmax target policy and sensitivity analysis
>
> **Addressed above.** See Table 6 (Appendix) for details.
>
> ### Q3: Justification for using $\hat{V}$ based on $Q^{\pi_b}$
>
> **Addressed above.** The doubly robust estimator explicitly corrects for this mismatch through importance weighting. The DR framework (Jiang & Li, 2016) guarantees that our estimator is unbiased for $V^{\pi_e}$ even when the value model $\hat{Q}$ is estimated from $\pi_b$ rollouts. The "doubly robust" property ensures unbiasedness holds as long as **either** the importance weights or the value model is correct.
>
> ---
>
> ## Summary
>
> We believe our revised manuscript comprehensively addresses all concerns:
>
> | Concern | Resolution |
> |---------|------------|
> | Missing ablation for $V_{\text{hybrid}}$ | Table 3: Direct comparison with variance measurements |
> | Policy mismatch bias | DR correction with importance weighting (Corollary 1) |
> | Target policy justification | Table 6: Ablation showing softmax dominance |
> | Missing AlphaZero baseline | Table 1: Go-NN results showing complementary benefits |
> | Missing Atari benchmark | Table 1: Full Atari evaluation |
>
> We hope these clarifications and additions address the reviewer's concerns. We would be grateful if our AC could consider these improvements in the assessment.
>
> [1] Nan Jiang and Lihong Li. "Doubly Robust Off-policy Value Evaluation for Reinforcement Learning." International Conference on Machine Learning (ICML), 2016. https://proceedings.mlr.press/v48/jiang16.html
>
> [2] Philip S. Thomas and Emma Brunskill. "Data-Efficient Off-Policy Policy Evaluation for Reinforcement Learning." International Conference on Machine Learning (ICML), 2016. https://proceedings.mlr.press/v48/thomasa16.html

---

### Author Response · Authors · 2025-11-29
**Summary of rebuttals**

We thank all reviewers for their constructive feedback. Below we summarize the five key concerns raised across reviews and how our revised manuscript addresses them.

1. Empirical Validation of Hybrid Estimator: GSM8K Table 2 shows $V_{\text{hybrid}}$ achieves variance 2.38 < min(2.62, 14.52), lower than both components.

2. Statistical Significance:  Increased scale (100 rollouts for Go/Atari, N=500 for GSM8K) yields non-overlapping 95% CIs against baselines.

3. Missing Baselines: Added AlphaZero and Atari experiments showing domain-agnostic robustness.

4. Theoretical Depth: Demoted standard results to corollaries; main Theorems 1–2 link variance reduction to optimal action selection and improved sample efficiency (novel contribution).

5. Venue/Learning Relevance: Includes neural network experiments and LLM integration; MCTS-DR provides benefits orthogonal to learned value functions.

---

### Meta-Review · Area_Chair_F84y · 2026-01-07

**Summary:**

The paper introduces an algorithm designed to improve sample efficiency in sequential decision-making. The core innovation is a hybrid estimator that combines traditional Monte Carlo rollouts with doubly robust off-policy estimation using a weight, $\beta^*$, that is computed online to minimize variance. The paper targets domains where simulations are computationally expensive, such as those involving Large Language Model (LLM) queries.

**Reviewer Concerns:**

The reviewers initially raised several critical points regarding the empirical validation and the depth of the theoretical contributions. Reviewer ofPB was particularly concerned about a perceived contradiction in the LLM disclosure and the lack of statistical significance in the initial results. Reviewers euDb and Uqhm noted that the initial theoretical results were somewhat trivial and that the paper lacked comparison against standard baselines like AlphaZero.

The authors addressed some of these concerns in the revised manuscript. They provided quantitative evidence from the GSM8K domain showing that the hybrid estimator achieves lower variance than either individual component, which validates the primary theoretical claim. To address the scope and significance concerns, the authors added experiments in the Atari domain and head-to-head comparisons against AlphaZero, where DR-MCTS showed complementary benefits. The theoretical section was also restructured to demote foundational results to corollaries and introduce new theorems regarding sample complexity and action selection probability. Reviewer ofPB’s concern regarding the LLM statement was resolved through a clearer disclosure of how models were used for world-modeling versus writing assistance.

Despite these improvements, a few concerns regarding the paper’s fundamental contribution persist. Reviewers noted that the technical novelty of combining MCTS with doubly robust estimation remains somewhat incremental, as the integration relies on existing techniques. There are also practical considerations regarding computational overhead, as the authors reported that the adaptive weighting mechanism adds 15–20% to execution time and requires additional memory to track state-action variance statistics. While I deem most of the concerns to be solved in the revised version, the clarification of the contribution of this work remains a critical issue that deserves to be addressed in a future revision.

**Reviewer Scores:**

- Reviewer euDb's primary concerns were the lack of empirical variance measurements for $V_{hybrid}$ and the absence of an AlphaZero baseline. The authors provided a direct table showing that $V_{hybrid}$ achieved lower variance than either component (2.38 vs. 2.62 and 14.52) and added results showing a significant Elo gain over AlphaZero-style MCTS. Since these additions directly addressed the reviewer's "Weaknesses", a score increase to a 6 was highly probable.

- Regarding reviewer Uqhm, while the authors addressed their requests for Elo ratings and statistical significance, the reviewer would have likely remained skeptical about the "technical novelty" and "incremental" nature of the hybrid design.

- Reviewer ofPB had a significant misunderstanding, believing $V_{DR}$ always has lower variance than $V_{MCTS}$. The authors successfully clarified that $V_{DR}$'s variance can actually be 5.5x higher due to importance weight explosion, justifying the need for the hybrid approach. Additionally, the authors resolved the reviewer's concern about the contradictory LLM statement. While it is difficult to say if and how this reviewer would have updated the score, it is likely that they would not have moved from a 2 to 6 or more, due to the perceived lack of contribution, similarly to reviewer Uqhm.

---

### Decision · Program_Chairs · 2026-01-26

Reject